

Genetic functional potential displays minor importance in explaining spatial variability of methane
fluxes within a *Eriophorum vaginatum* dominated Swedish peatland.
Joel D. White [a, *], Lena Ström [a], Veiko Lehsten [a,c], Janne Rinne [a,d], Dag Ahrén [b]
[a] Department of Physical Geography and Ecosystem Science, Lund University, Sölvegatan 12 S-223 62 Lund,
Sweden. joel.white@nateko.lu.se; lena.strom@nateko.lu.se; veiko.lehsten@nateko.lu.se;
janne.rinne@nateko.lu.se
[b] National Bioinformatics Infrastructure Sweden (NBIS), Department of Biology, Sölvegatan 35 Lund
University, 22362 Lund, Sweden. dag.ahen@biol.lu.se
[c] Swiss Federal Institute for Forest, Snow and Landscape research (WSL), Birmensdorf, Switzerland.
[d] Natural Resources Institute Finland, Production Systems, Latokartanonkaari 9, 00790 Helsinki, Finland
Correspondence: Joel D. White (joel.white@nateko.lu.se)



**Abstract.** Microbial communities of methane ($CH_4$) producing methanogens and consuming methanotrophs play
an important role for Earth's atmospheric $CH_4$ budget. Despite their global significance, knowledge on how much
they control the spatial variation in $CH_4$ fluxes from peatlands is poorly understood. We studied variation in $CH_4$
producing and consuming communities in a natural peatland dominated by *Eriophorum vaginatum*, via a
metagenomics approach using custom designed hybridization-based oligonucleotide probes to focus on taxa and
functions associated with methane cycling. We hypothesized that sites with different magnitudes of methane flux
are occupied by structurally and functionally different microbial communities, despite the dominance of a single
vascular plant species. To investigate this, nine plant-peat mesocosms dominated by the sedge *Eriophorum*
*vaginatum*, with varying vegetation coverage, were collected from a temperate natural wetland and subjected to a
simulated growing season. During the simulated growing season, measurements of $CH_4$ emission, carbon dioxide
($CO_2$) exchange and $\delta^{13}C$ signature of emitted $CH_4$ were made. Mesocosms 1 through 9 were classified into three
categories according to the magnitude of $CH_4$ flux. Gross primary production and ecosystem respiration followed
the same pattern as $CH_4$ fluxes, but this trend was not observed in net ecosystem exchange. We observed that
genetic functional potential was of minor importance in explaining spatial variability of $CH_4$ fluxes with only
small shifts in taxonomic community and functional genes. In addition, a higher $\beta$-diversity was observed in
samples with high $CH_4$ emission. Among methanogens, *Methanoregula,* made up over 50% of the community
composition. This, in combination with the remaining hydrogenotrophic methanogens matched the $\delta^{13}C$ isotopic
signature of emitted $CH_4$. However, the presence of acetoclastic and methylotrophic taxa and type I, II and
*Verrucomicrobia* methanotrophs indicates that the microbial community holds the ability to produce and consume
$CH_4$ in multiple ways. This is important in terms of future climate scenarios, where peatlands are expected to alter
in nutrient status, hydrology, and peat biochemistry. Due to the high functional potential, we expect the
community to be highly adaptive to future climate scenarios.





## 1.0 Introduction

Methane (CH$_4$) is second most important long-lived greenhouse has in the atmosphere (Dean et al., 2018; Dlugokencky et al., 2009; Saunois et al., 2020). Atmospheric CH$_4$ concentrations have increased twofold during the industrial period (Dlugokencky et al., 2003). Following a decade of near-zero increase by the turn of the millennium, globally averaged atmospheric CH$_4$ is on the rise again at a rate of 5 ppb yr$^{-1}$ (Dlugokencky et al., 2009; Dlugokencky et al., 2003; Saunois et al., 2016b; Saunois et al., 2020). CH$_4$ is emitted from both natural and anthropogenic sources (Dean et al., 2018; Saunois et al., 2016a). Within all natural sources, the largest contributor to CH$_4$ emissions are wetlands producing 149Tg (range 102–182) CH$_4$ yr$^{-1}$, i.e. 40 % of the total natural CH$_4$ emission (Dean et al., 2018; Saunois et al., 2020).

Microbial CH$_4$ emission is a byproduct of microbial metabolism and is produced by methanogenic *Archaea* following hydrolysis and fermentation (Ferry, 1999). CH$_4$ production occurs under anoxic conditions, where organic carbon bound to dead organic matter is converted into CH$_4$ via methanogenesis (Ferry, 1999). Methanogenesis is the final reaction in anaerobic degradation of organic matter and occurs stepwise in cooperation between different microbial functional groups.

Methanotrophs, of the phyla, *Proteobacteria*, *Verrucomicrobia*, and candidate phylum NC10 act as a natural bio-filter by oxidizing CH$_4$ and thereby reducing emissions. Inhabiting the oxic-anoxic interfaces, methanotrophs oxidize between 10 to 90% of the CH$_4$ produced by methanogenic archaea before it is emitted to the atmosphere (Hakobyan and Liesack, 2020; Wendlandt et al., 2010). Methanotrophs can be found in a number of environments including wetlands, marine or freshwater sediments, rice paddies and sewage, and grow on one-carbon compounds such as methanol and methylated amines (Wendlandt et al., 2010; Chen et al., 2008; Dedysh, 2002, 2009). A common characteristic of all aerobic methanotrophs is their ability to oxidize CH$_4$ to carbon dioxide (CO$_2$) and water.

CH$_4$ emissions from natural wetlands are known to exhibit both spatial and temporal variability (Crill et al., 1988; Sun et al., 2013). The spatial variability makes wetland CH$_4$ emissions difficult to model and predict (Wania et al., 2009, 2010), as CH$_4$ emission within similar environmental conditions (i.e. ecotype) can vary by several orders of magnitude without an apparent explanation (Bridgham et al., 2013). According to current knowledge, both production and consumption of CH$_4$ within peatland ecotypes is driven by (i) water table depth (WTD), which determines the thickness of oxic and anoxic zones; (ii) plant species composition, which provides substrates and plant mediated transport of CH$_4$ to the atmosphere; (iii) soil temperature, which affects the rate of microbiological



processes; and (iv) substrate availability for biogeochemical processes such as methanogenesis and
methanotrophy (Joabsson et al., 1999; Korrensalo et al., 2018; Mastepanov et al., 2013; Strack et al., 2004; Ström
et al., 2015).
Recent advancements in molecular techniques have allowed researchers to explore further drivers affecting the
magnitude of $CH_4$ fluxes (Fierer et al., 2014; Galand et al., 2003; Galand et al., 2002; Juottonen et al., 2008).
Shifts within microbial community composition and function, where metabolic processes occur, are expected to
contribute to the observed variability of $CH_4$ fluxes within ecotypes (Bridgham et al., 2013; Dean et al., 2018).
Therefore, the ability to include the functional potential of microbial communities as a potential driver of $CH_4$
fluxes has gained more attention.
The field of environmental genomics has developed rapidly, utilizing the genetic material taken from un-cultured
environmental samples to identify accurately the functional gene composition (Ungerer et al., 2008; Ward et al.,
2008). Techniques include the establishment of polymerase chain reaction (PCR) based studies, where marker
genes are used to evaluate microbial community composition via amplification of regions conserved across
species (Brumfield et al., 2020; Lane et al., 1986). The targeting of the marker gene 16S in ribosomal ribonucleic
acid (rRNA) that occur in *Bacterial* and *Archaea* genomes has often been recognized as the gold standard in
prokaryotic identification (Brumfield et al., 2020; Lane et al., 1986). In $CH_4$ research, key genes such as methyl
coenzyme M reductase (*mcrA*) and methane monooxygenase component A alpha chain (*mmoX*) are often targeted
to determine community composition and functional potential (Chroňáková et al., 2019; Freitag et al., 2010;
Galand et al., 2005; Liebner et al., 2012). However, recent research has suggested that studying the entire
metagenome increases the possibility to predict soil functional potential as opposed to enriching for singular genes
(Gravel et al., 2012; Kushwaha et al., 2015; Manoharan et al., 2015).
In order to attain the necessary depth of sequencing coverage required to analyze the functional potential of soil
microbial communities, whole metagenomic sequencing is required (Dinsdale et al., 2008; Fierer et al., 2014).
Though, even with the constant advancements in sequencing technology, metagenomics studies require large
financial and computational resources to obtain the necessary depth of coverage to ensure small microbial
communities, including *Archaea*, are detected (Escobar-Zepeda et al., 2015; Pereira-Marques et al., 2019). In
response to these limitations, we applied the molecular technique "captured metagenomics", which targets key
genes related to the metabolism of both methanogenic *Archaea* and methanotrophic *Bacteria* (Kushwaha et al.,
2015; Manoharan et al., 2015).



Captured metagenomics provides an alternative to studying the entire deoxyribonucleic acid (DNA) pool of
metagenomic communities (Gasc et al., 2016; Kushwaha et al., 2015; Manoharan et al., 2015). The sequence
capture technique hybridizes DNA fragment targets from a metagenomic DNA fragmented pool through the
custom set of probes designed via the MetCap pipeline (Kushwaha et al., 2015). This method makes it possible to
target thousands of key genes related to methanogen and methanotroph metabolism, while avoiding lengthy lab
hours and massive sequencing efforts required of large-scale metagenomic study. In addition, this allows for
multiple biological replicates at a reasonable cost per sample (Gasc et al., 2016; Kushwaha et al., 2015; Manoharan
et al., 2015).
Here, we address the functional potential impact of $CH_4$ producing and consuming microbes on the magnitude of
$CH_4$ flux. To determine the functional genetic diversity, we apply captured metagenomics on genes encoding for
enzymes related to $CH_4$ metabolism on nine peat-plant mesocosms dominated by the sedge *Eriophorum*
*vaginatum*. We aim to (1) identify whether the composition of both $CH_4$ producing and consuming taxa shift in
dissimilarity in response to variations in $CH_4$ flux, (2) determine whether the $\beta$-diversity increases with increasing
$CH_4$ emission; and finally, (3) identify whether the $\delta^{13}C$ of emitted $CH_4$ matches the dominant taxa in samples.
**2.0 Methodology**
**2.1 Site description**
To study the functional diversity of a microbial community producing and consuming $CH_4$, we collected peat-
plant mesocosms from Fäjemyr, an ombrotrophic bog located in Skåne, southern Sweden (56°15'53.3"N
13°33'14.1"E). The peatland is classified as an eccentric bog, and is dominated by semi-forested areas alternating
between raised hummocks, hollows and moss lawns (Lonnstad and Löfroth, 1994; Lund et al., 2007). Long-term
(1961-1990) mean annual temperature and precipitation are 6.2°C and 700mm respectively (Smhi, 2006). The
peat depth ranges between 4-5m, while the peat water pH is generally below 4 throughout the entirety of the
growing season (Lund et al., 2007).
Vegetation composition at Fäjemyr is diverse including hummocks dominated by dwarf shrubs such as *Calluna*
*vulgaris* and *Erica tetralix*. The moss lawns are carpeted with *Sphagnum*-mosses including *S. magellanicum* and
*S. rubellum*, while the raised drier hummocks are dominated by dwarf Scots pine (*Pinus sylvestris*). The dominant
sedge species within the site is *Eriophorum vaginatum* (Lonnstad and Löfroth, 1994; Lund et al., 2007).



**2.2 Experimental design**
A total of 9 cylindrical mesocosms (height: 26 cm, diameter: 27 cm) were collected on the 30[th] of March, 2017.
The mesocosms, numbered M1-M9, were carefully cut from the peatland, transferred directly into plastic
containers for transportation to Lund University (82km away) where they were incubated under temperature and
light controlled conditions in a growth room. Over the first month we started at 10 ºC and no light, temperature
and light levels were gradually increased to allow the mesocosms to adjust and to simulate the onset of the growing
season. For the final 4 weeks of the experiment, the conditions in the growth room were kept at 20 °C, 500 µmol
PAR m$^{-2}$ s$^{-1}$ and 17 daylight hours (based on sunrise and sunset at Fäjemyr). Due to the effect of heat radiation
from the lamps, the temperature varied over the day from 18±0.3 °C when the lamps were off to 23±1 °C when
they were on. Additionally, the light level varied (512±42 µmol PAR m$^{-2}$ s$^{-1}$) somewhat over the surface due to
variations in individual lamp efficiency. The mesocosms were rotated bi-weekly to minimize the effect of spatial
variations in growth conditions.
All mesocosms were watered daily with deionized water to maintain a constant water table depth at 5 cm below
the surface. During the experiment, weekly to bi-weekly (final 3 weeks, n = 6) measurements of $CO_2$ and $CH_4$
fluxes were conducted. The $\delta^{13}C$ of emitted $CH_4$ was measured on three occasions in the final weeks. Upon
completion of the experiment, peat samples were removed from the top oxic-anoxic interface (5 cm), bottom (15
cm) and from the peat sticking to the root surface (rhizosphere) for DNA extraction. Resulting in a total of 27
samples peat samples for genomic analysis (each mesocosm n = 3)
**2.3 Flux measurements**
Flux measurements of $CO_2$ and $CH_4$ were made using the static chamber technique (Crill et al., 1988; Livingston
and Hutchinson, 1995). For each mesocosm, 6-minute-long measurements in both light and dark conditions were
conducted to establish Net Ecosystem Exchange (NEE) and Ecosystem Respiration ($R_{eco}$). We used a negative
sign convention where negative values indicate an uptake of $CO_2$ from the atmosphere and positive a release.
Gross Primary Production (GPP) was calculated according to the relationship GPP = NEE - $R_{eco}$. Measurements
were performed using a transparent 5-liter cylindrical polycarbonate chamber that was covered with a dark hood
for $R_{eco}$ measurements. The chamber was equipped with a rubber list to ensure an airtight seal and a fan to circulate
air. Both $CO_2$ and $CH_4$ concentrations were measured with a LGR Fast Greenhouse Gas analyser (model 911-
0010, Los Gatos Research, CA USA). The $CO_2$ and $CH_4$ fluxes were calculated via changes in gas concentration



as a function of time using linear fitting over 6-minute measurement periods. Data was corrected for both air
pressure, volume of the chamber and ambient air temperature.
**2.4 Stable isotope analysis**
The $CH_4$ emission and its $\delta^{13}C$ signature were determined using a cavity ring-down laser absorption spectrometer
with the closed chamber technique described above (G2201i, Picarro, Santa Clara, USA). The surface of each
peat mesocosm was covered with a transparent cylindrical chamber for 25-30 minutes while the $CH_4$ mixing ratio
and $\delta^{13}C$-$CH_4$ was recorded with 1 second intervals. Data was averaged into one minute averages. $CH_4$ emission
were calculated using linear fitting, and the $\delta^{13}C$ signature of emitted $CH_4$ was determined with a Keeling plot
intercept approach (Keeling, 1958; Thom et al., 1993). The resulting $\delta^{13}C$-$CH_4$ values were corrected by adding a
constant value of 3.4 ‰, based of comparison with isotopic mass spectrometer.
**2.5 Captured metagenomics**
**2.5.1 Peat samples and DNA extraction**
Peat material was collected from three sampling locations within each mesocosm. Samples were taken from the
top oxic-anoxic interface (5 cm), bottom (15 cm) and from the root adjacent peat directly attached to the root
surface (10 cm). The peat material was stored at 20°C and then thawed at 4°C prior to DNA extraction. DNA was
extracted following the DNeasy® PowerSoil® Kit (Qiagen, Hilden, Germany) and carried out according to the
manufacturer's protocol, following the recommended 0.25 g of input material. After DNA extraction, samples
were tested for quality (absorbance ratio 260/280) and concentration on a NanoDrop lite (NanoDrop Technologies,
Willington NC, USA) and Invitrogen Qubit 4 fluorometer (Thermo Fisher Scientific, Waltham MA, USA)
respectively.
**2.5.2 SeqCap EZ probe generation**
Genes encoding enzymes closely related to the $CH_4$ production and oxidation in pathway map00680 were
identified from the Kyoto Encyclopedia of Genes and Genomes (KEGG). The nucleotide sequences were
downloaded via a custom R script (https://github.com/dagahren/metagenomic-project). In total, 548,104 genes
were downloaded and compiled into a local database, subsequently referred to here as the $CH_4$ database. The
nucleotide coding sequences of the $CH_4$ database were used to design hybridisation-based probes for sequence
capture. The probe sequences were generated using the MetCap pipeline, where sequences were clustered with





90% sequence similarity with an average of 4 probes per cluster (Kushwaha et al., 2015; Manoharan et al., 2015).
In total, 193,386 individual probes were generated after clustering. They were generated with a melting
temperature of 55°C and probe length 40mer which is suitable for use with our protocol that is based on
NimbleGen SeqCap EZ (Roche NimbleGen Inc., Madison, USA).

**2.5.3 Probe hybridisation, library generation and sequencing**

Depending on the extracted DNA concentration, 150 ng or 1 µg of genomic DNA in a total volume of 100 µl low
TE, was sheared for 13 cycles of 30s on, 30 s off, using a Bioruptor Pico and 0.65 ml Bioruptor tubes (Diagenode
SA, Seraing, Belgium). 1 µl of the sheared samples was run on a DNA HS chip prior to contamination clean up.
The fragmented DNA was then purified using 1.8× AMPure XP beads (Beckman Coulter, Indianapolis, USA)
and used as input material for preparation of pre-capture libraries and constructed according to the Nimblegen
SeqCap EZ HyperCap Workflow User's Guide (Version 1.0, June 2016). We used two modifications to this
method to allow for improved hybridization: (i) for the adapter ligation step, 5 µl of 15 µM KAPA unique dual
index mixed adapters were used instead of single index adapters, (ii) for the pre-capture PCR, 7 cycles was used
for libraries with a genomic DNA input of 150 ng, and 5 cycles where the input was 1 µg.
Pre-capture libraries were purified with 1.8x Ampure beads and quantified by Quant-iT double-stranded DNA
high sensitivity assay and the average fragment size determined by analysis on a Fragment Analyser (Agilent,
Santa Clara, USA) using a high sensitivity NGS Kit. Libraries were multiplexed in pools of 15 in equimolar
amounts based on the aforementioned concentrations and sizes. 1 µg of each pool was transferred to a test tube
and hybridised to custom probes according to the NimbleGen SeqCap EZ SR User's Guide (Version 4.3, October
2014). When setting up the hybridisations, SeqCap EZ Developer Reagent and HyperCap Universal Blocking
Oligos were added to each pool, according to manufacturer's instructions. The capture tubes were incubated in a
thermal cycler set at 47 °C, with the heated lid set to 57 °C for 69 hours.
The final captured library pool was reagent-treated and further purified using 1.8× AMPure XP beads to remove
unligated adapters. The quantity and quality of the final pool was assessed by Qubit and Bioanalyzer and
subsequently by qPCR using the Illumina Library Quantification Kit from Kapa on a Roche Light Cycler
(LC480II, Basel, Switzerland). Briefly, a 20 µl PCR reaction (performed in triplicate for each pooled library) was
prepared on ice with 12 µl SYBR Green I Master Mix and 4 µl diluted pooled DNA (1:1000 to 1:100,000
depending on the initial concentration determined by the Qubit). PCR thermal cycling conditions consisted of



initial denaturation at 95°C for 5 minutes, 35 cycles of 95°C for 30 seconds (denaturation) and 60°C for 45 s for
annealing and extension, a melt curve analysis at 95°C and cooling at 37°C.
The captured libraries were sequenced on an Illumina HiSeq4000 platform using sequencing by synthesis
technology to generate 2 x 150 base pair (bp) paired-end reads, the analysis was carried out by the Centre for
Genomic Research, University of Liverpool, United Kingdom.
**2.6 Data Analysis**
Raw fastq files were trimmed for the presence of Illumina adapter sequences using Cutadapt version1.2.1 (Martin,
2011). The option -O 3 was used, which means that the 3' end of any reads that matched the adapter sequence
were removed by 3bp. The reads were further trimmed using Sickle version 1.200 with a minimum window quality
score of 20 (Joshi, 2011). This meant that reads shorter than 20bp were removed.
Following sequence trimming, reads from each of the captured data sets were submitted to Metagenomic Rapid
Annotations using Subsystems Technology (MG-RAST) for sequence annotation (Meyer et al., 2008). Default
parameters were used for quality filtering of bad reads and removal of sequence duplicates. Once annotated,
sequences were filtered for both taxonomic and functional annotations via the KEGG $CH_4$ metabolism filter
(ko:00680). The taxonomic and functional annotations from MG-RAST were annotated using refseq and KEGG
(KO) databases (Kanehisa et al., 2015; O'leary et al., 2016). and exported to R for downstream analysis.
**2.7 Statistical analysis**
All statistics were completed in R and visualized using 'ggplot2' (Hadley, 2016). Given the small sample size (n
= 9), as well as the non-normal distribution of the values, a permutation test was used based around the 6 temporal
replicates (M1-9 n = 6, total n = 54) of $CH_4$ and $CO_2$ flux from each mesocosm and we used an 'independence
test' in R from the 'coin' package (Hothorn et al., 2021). We tested pairwise for differences in means and
performed a subsequent correction for multiple testing as described by Holm (1979). To evaluate the statistical
relationship between $CH_4$ and $CO_2$ flux, a Pearson's correlation test was used from the package 'corrplot' (Wei
and Simko, 2017).
Further statistical tests for use on genomic data, including the Permutational multivariate analysis of variance
(PERMANOVA), α-diversity and β-diversity, and Nonmetric Multidimensional Scaling (NMDS), were
completed using the 'vegan' package (Oksanen et al., 2019). Due to the low number of replicates and non-normal



distribution we performed a PERMANOVA to determine the most influential taxa and functional genes
(Anderson, 2001). Input data for the PERMANOVA was double root transformed to reduce the influence of highly
abundant taxa and genes. When computing the PERMANOVA and NMDS Bray-Curtis distances were used to
quantify the compositional dissimilarity between groups with 999 permutations (Anderson, 2001). To test for
significance between flux categories, we performed a pairwise comparisons between group levels with False
Discovery Rate (FDR) corrections for multiple testing via the 'RVAideMemoire' package (Herv, 2021). Finally,
the similarity percentage test (SIMPER) was used to evaluate the contribution of individual taxa and genes to the
overall Bray-Curtis dissimilarity, a cut off of 70% was applied (Warton et al., 2012).
**3.0 Results**
**3.1 Mesocosm characteristics**
**3.1.1 Carbon fluxes**
Carbon fluxes of $CH_4$ and $CO_2$ vary among the mesocosms and are shown in fig 1. The mean flux of $CH_4$
mesocosms ranged between 152 (SD ±54) in M9 to 371 (SD ±23) $\mu$mol m$^{-2}$ h$^{-1}$ in M4. After observing such large
variability within $CH_4$ fluxes, we performed a pairwise randomization and established that M4 had a significantly
higher flux than M1-3 and M5-9 ($p \leq 0.0001$) while M9 had a significantly lower flux ($p \leq 0.0005$) than the
remaining mesocosms. For further analysis, we separated the measurements from M4 and M9 from the remaining
mesocosms, enabling us to test for a plausible explanation to the observed differences between fluxes when
compared to the structure and function of the microbial community. Hereafter, M4 will be referred to as HFM
(high flux mesocosm), M9 as LMF (low flux mesocosm) and the remaining mesocosms as MFM (medium flux
mesocosm).
GPP and $R_{eco}$ generally followed the same observed pattern as $CH_4$. With HFM being significantly higher than
MFM ($p \leq 0.008$) and LFM ($p \leq 0.004$) in GPP. While $R_{eco}$ was also significantly different between HFM - MFM
($p \leq 0.001$) and HFM - LFM ($p \leq 0.002$). However, the same trend was not observed in NEE, where the highest
recorded mean flux was observed in M8 (i.e. MFM category), not in the HFM category as observed in GPP and
$R_{eco}$.



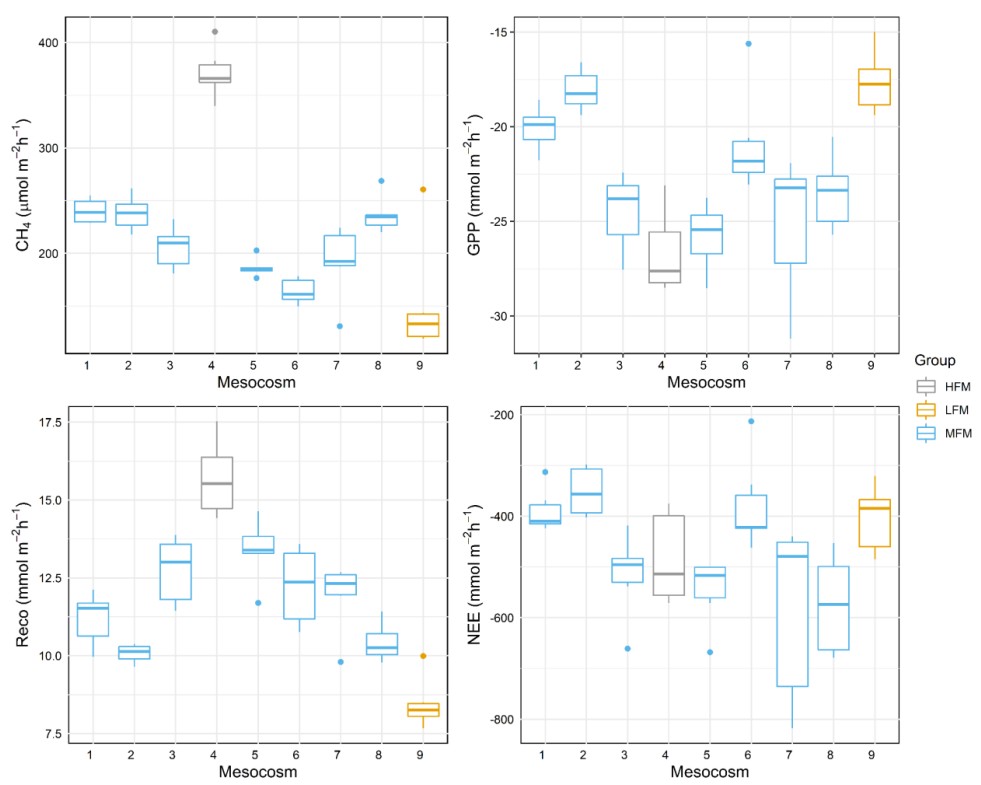


**Figure 1:** Boxplots of carbon fluxes measured during the last 6 weeks of the lab experiment. The boxes
show quartiles and the median, the whiskers denote data within 1.5 times of the interquartile range and
the closed circles denote outliers. Methane flux ($CH_4$), Gross Primary Productivity (GPP), Ecosystem
Respiration ($R_{eco}$), and Net Ecosystem Exchange (NEE). Note the units on the y-axis (mesocosms 1 –
9: n = 6)

In an attempt to investigate the relationships between carbon fluxes we conducted a correlation test and found that

the flux of $CH_4$ held a positive relationship to $R_{eco}$ ($R^2 = 0.60$, $p \leq 0.04$), but not to GPP or NEE (fig 2). When

analysing $CO_2$ fluxes, GPP held a strong negative relationship to $R_{eco}$ ($R^2 = 0.70$, $p \leq 0.002$), while NEE held a

strong positive relationship to GPP ($R^2 = 0.82$, $p \leq 0.001$) (fig 2).

**3.1.2 Vegetation**

The peatland mesocosms were dominated by the sedge *E. vaginatum*, but also included small amounts of the

*Sphagnum*-mosses *S. magellanicum* and *S. rubellum*. The number of sedge tillers ranged between 384 in HFM,

(mean) in MFM and 134 in LFM (fig 2). The number of *E. vaginatum* tillers held a strong correlation

coefficient and significant relationship to GPP ($R^2 = 0.95$, $p \leq 0.01$) and $R_{eco}$ ($R^2 = 0.94$, $p \leq 0.01$) (fig 2). While





the remaining carbon fluxes $CH_4$ ($R^2$ = 0.44, p > 0.05) and NEE ($R^2$ = 0.64, p > 0.05) had a high correlation
coefficient, this relationship was not significant to the number of *E. vaginatum* tillers.

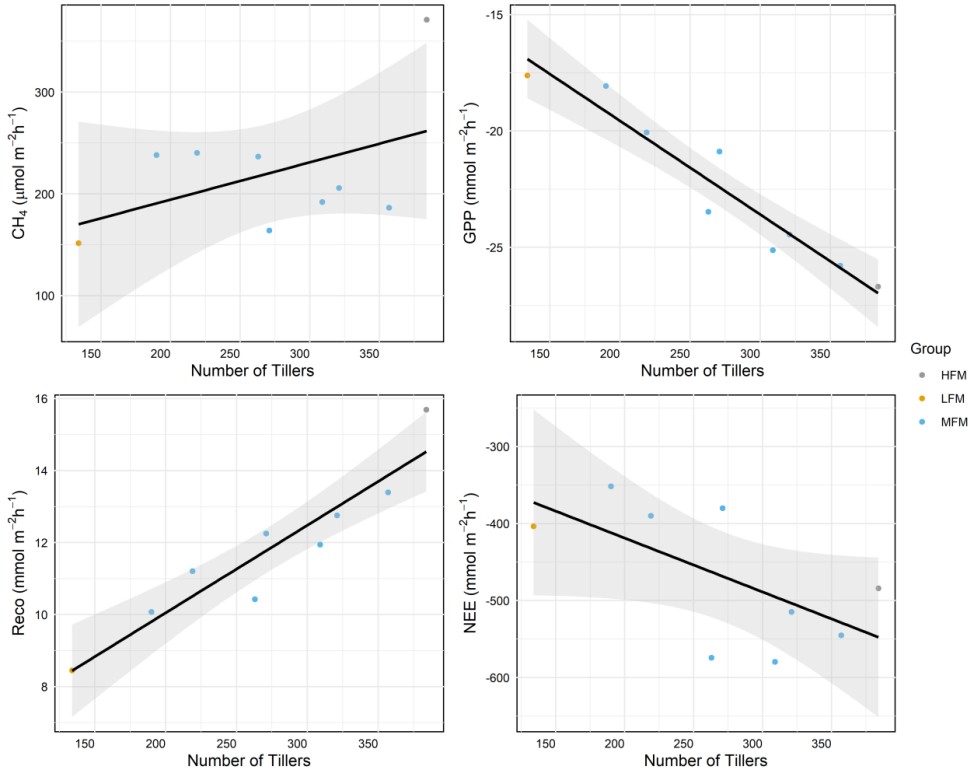


**Figure 2:** The relationship between mesocosm carbon fluxes and the number of tillers of *E. vaginatum*.
Data points represent the mean flux of each individual mesocosm while the shaded area indicates the
95% confidence level interval for predictions of the linear model.
**3.1.3 Isotopic signature**
Distinct isotopic signatures of individual mesocosms are shown in fig 3. All mesocosms fell within the range of
hydrogenotrophic methanogenesis ($\delta^{13}C$ = −110‰ to −60‰) (Chanton, 2005; Whiticar, 1999). However, M2
(MFM) and M4 (HFM) indicated a slight tendency towards acetoclastic methanogenesis with less negative
isotopic signature ($\delta^{13}C$ = -60‰ to -50‰), both yielding mid -60‰ $\delta^{13}C$ Keeling intercepts. A significant positive
correlation ($R^2$ = 0.5, p ≤ 0.001) and significant relationship also existed between $CH_4$ flux and the Keeling
intercept shown in fig 3.



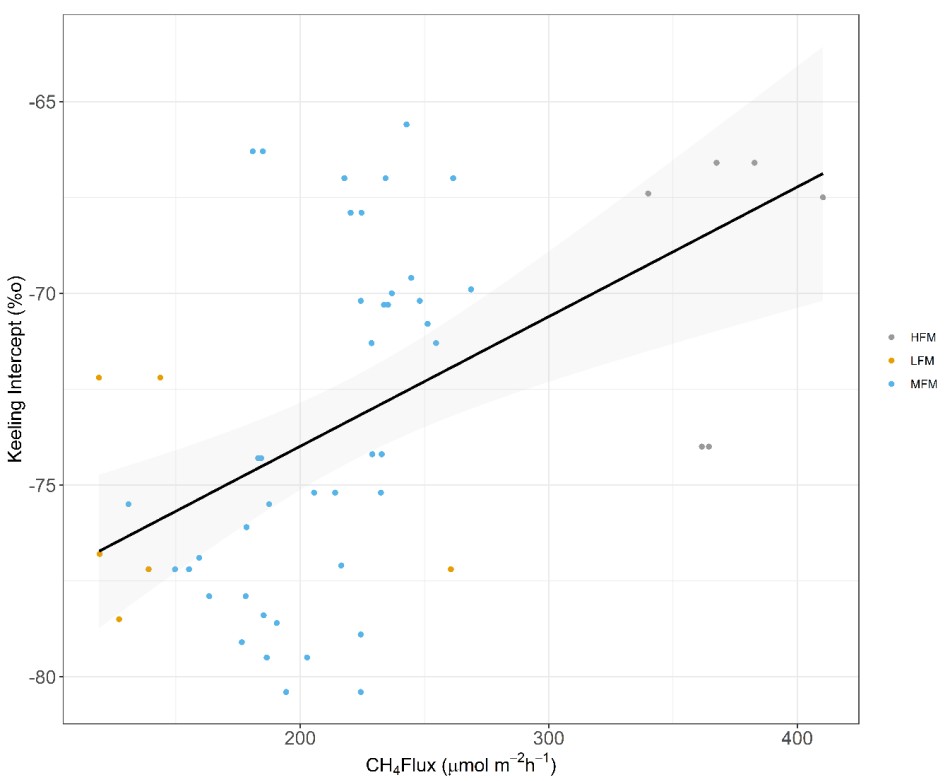


**Figure 3:** Scatter plot visualizing the relationship between $CH_4$ flux ($\mu mol\ m^{-2}\ h^{-1}$) and Isotopic signature
of emitted $CH_4$ (keeling intercept ‰). Data points colored by flux category while the shaded area
indicates the 95% confidence level interval for predictions of the linear model.

**3.2 Captured Metagenomics**

**3.2.1 Microbial community composition**

Diverse methanogenic *Archaea* and methanotrophic *Bacteria* were observed in all samples. In total, 20

methanogenic *Archaea* and 5 methanotrophic *Bacteria* were detected. Methanogens which utilize $CO_2 + H_2$,

methanol, acetate and methyl amines substrates for ATP and biomass production were all observed throughout

the samples, which indicates a high functional potential. Although less diverse than the methanogens,

methanotrophs from *Alphaproteobacteria*, *Gammaproteobacteria* and *Verrucomicrobia Phylum* were present in

all samples. However, due to the environmental conditions no methanotrophs from the NC10 *Phylum* were

detected. This community composition resulted in a median $\alpha$-diversity measure of 2.38, which is a measure of

the diversity of the peatland ecosystem.



### 3.2.2 Total taxonomic distribution

At genus level, 20 methanogenic genera were identified. The highest relative abundance of methanogens included *Methanoregula* which contributed 54% to the total proportion, followed by *Methanosarcina* (17%), *Methanosphaerula* and *Methanothermobacter* which contributed 5% each to the total proportion of methanogens (fig 2). Within the methanogen community, genera with the ability to metabolize via hydrogenotrophic, acetoclastic and methylotrophic methanogenesis pathways were also detected. Hydrogenotrophic methanogens made up (78%) of the total proportion, while *Methanosarcina* which can utilize several substrates for ATP and biomass production contributed 17%, followed by methylotrophic methanogens (<5%) and finally, acetoclastic methanogens which contributed to <1% of the total.

In addition to methanogens, 5 genera of $CH_4$ reducing *Bacteria* were detected including methanotrophs from *Alphaproteobacteria, Gammaproteobacteria* and *Verrucomicrobia* class. Type II *Alphaproteobacteria* was the dominant *Subphylum,* including both *Methylocella* (37%) which contributed to largest proportion, followed by *Methylosinus* (28%). Type I *Gammaproteobacteria* genera *Methylococcus* (14%) and *Methylobacter* (10%) represented the lowest proportion. Finally, *Verrucomicrobia* included one genus, *Methylacidiphilum,* which contributed to 10% of the total proportion of methanotrophs.

### 3.2.3 β-diversity

β-diversity, which measures the change in diversity of species from one category to another, was measured as mean distance to the group centroid and highest in HFM (fig 4). HFM resulted in an average distance to median of 0.046, followed by MFM (0.042) and LFM (0.031). The largest distance between medians to centroids was observed between HFM and LFM, while the smallest distance between medians to centroids was observed between MFM and LFM. Due to a high variation and lack of replication, this relationship was observed as non-significant. Although the values for β-diversity are low, the differences between centroids indicates that communities of methanogens and methanotrophs become more similar to each other as the magnitude of flux decreases.





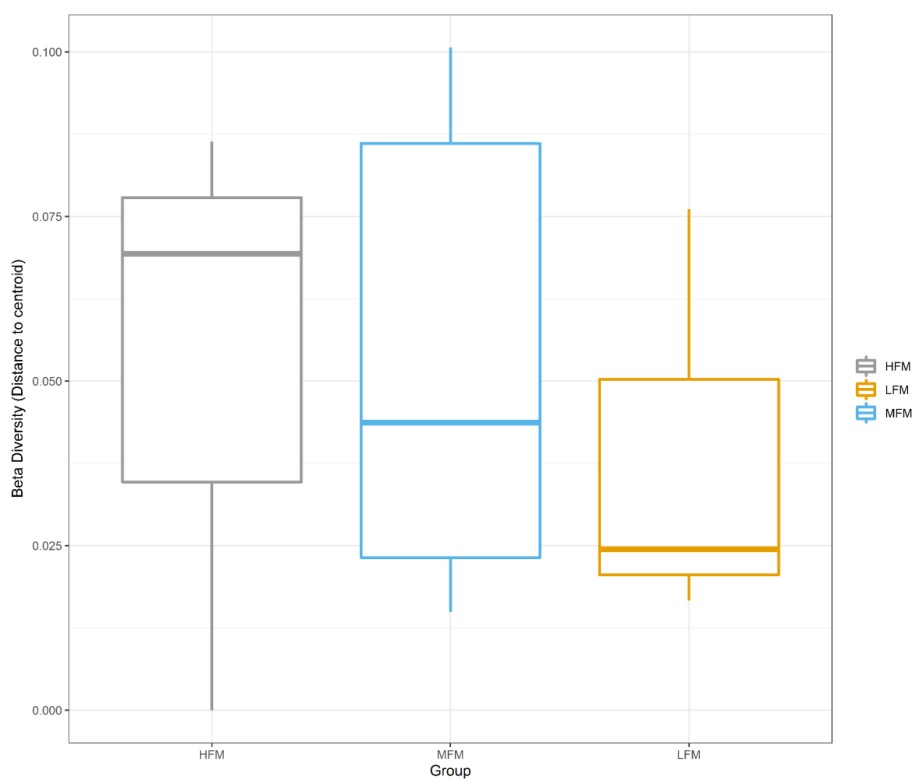

330

**Figure 4**: β-diversity boxplot of multivariate homogeneity of groups using Bray-Curtis distances. Dispersions of samples analyzed at genus level across HFM, MFM and LFM (HFM n = 3, MFM n = 21, LFM n = 3).

### 3.2.4 Taxonomy

The PERMANOVA and SIMPER analysis showed that the variation between the relative abundance of taxa between HFM, MFM and LFM was not significant. The small differences resulted in a non-significant weak correlation where 6% of the variation in taxa could be explained by HFM, MFM or LFM ($R^2 = 0.06$, $p \geq 0.05$). When comparing the relative abundance of methanogens and methanotrophs between HFM, MFM and LFM, five taxa including *Methanoregula*, *Methanosarcina*, *Methylocella*, *Methylosinus* and *Methylobacter* always contributed to the top 70% of cumulative sums (table 1, 2 and 3). However, in the HFM to LFM comparison, the addition of a sixth genus, *Methylacidiphilum,* was required to reach the 70% cut off (Table 3).

In all three comparisons, we observe that the hydrogenotrophic *Methanoregula* contributed the most to dissimilarity (table 1, 2 and 3) while type II *Alphaproteobacteria genera, Methylocella* and *Methylosinus*





contributed the second and third highest between flux categories. The order of contributions from the remaining
taxa *Methanosarcina*, *Methylobacter* and *Methylacidiphilum* changed depending on the comparison between
HFM, MFM and LFM.
**Table 1:** Results of SIMPER analysis. Taxa are ranked according to their average contribution to
dissimilarity between MFM and HFM. Average abundances, ratio (between averages using the greatest
common denominator), relative contribution of taxa and *p*-value of the permutation test (Probability of
getting a larger or equal average contribution in random permutation of the group factor) are also
included. A cut-off at a cumulative dissimilarity of 70% was applied.

| Genus | Average | SD | Avg. MFM | Avg. HFM | Ratio | Relative Contribution (%) | p - value |
|---|---|---|---|---|---|---|---|
| *Methanoregula* | 0.094 | 0.06 | 4370 | 7752 | 115:204 | 33% | 0.17 |
| *Methylocella* | 0.035 | 0.02 | 4826 | 5842 | 19:23 | 12% | 0.39 |
| *Methylosinus* | 0.035 | 0.02 | 3440 | 4586 | 1720:2293 | 13% | 0.21 |
| *Methanosarcina* | 0.019 | 0.01 | 1985 | 2699 | 1985:2699 | 7% | 0.31 |
| *Methylobacter* | 0.015 | 0.01 | 1234 | 1810 | 617:905 | 5% | 0.26 |


**Table 2:** Results of SIMPER analysis. Taxa are ranked according to their average contribution to
dissimilarity between MFM and LFM. Average abundances, ratio (between averages using the greatest
common denominator), relative contribution of taxa and *p*-value of the permutation test (Probability of
getting a larger or equal average contribution in random permutation of the group factor) are also
included. A cut-off at a cumulative dissimilarity of 70% was applied.

| Genus | Average | SD | Avg. MFM | Avg. LFM | Ratio | Relative Contribution (%) | p - value |
|---|---|---|---|---|---|---|---|
| *Methanoregula* | 0.061 | 0.04 | 4370 | 3757 | 4370:3757 | 26% | 0.91 |
| *Methylocella* | 0.033 | 0.02 | 4826 | 5111 | 254:269 | 15% | 0.45 |
| *Methylosinus* | 0.032 | 0.02 | 3440 | 4323 | 3440:4323 | 14% | 0.32 |
| *Methylobacter* | 0.017 | 0.01 | 1234 | 1647 | 1234:1647 | 8% | 0.10 |





| *Methanosarcina* | 0.015 | 0.01 | 1234 | 1647 | 1234:1647 | 7% | 0.95 |
| --- | --- | --- | --- | --- | --- | --- | --- |


**Table 3:** Results of SIMPER analysis. Taxa are ranked according to their average contribution to
dissimilarity between HFM and LFM. Average abundances, ratio (between averages using the greatest
common denominator), relative contribution of taxa and *p*-value of the permutation test (Probability of
getting a larger or equal average contribution in random permutation of the group factor) are also
included. A cut-off at a cumulative dissimilarity of 70% was applied.

| Genus | Average | SD | Avg. HFM | Avg. LFM | Ratio | Relative Contribution (%) | p - value |
| --- | --- | --- | --- | --- | --- | --- | --- |
| *Methanoregula* | 0.063 | 0.07 | 7752 | 3757 | 456:221 | 26% | 0.71 |
| *Methylocella* | 0.035 | 0.03 | 5842 | 5111 | 5842:5111 | 15% | 0.44 |
| *Methylosinus* | 0.034 | 0.02 | 4586 | 4323 | 4586:4323 | 15% | 0.32 |
| *Methanosarcina* | 0.017 | 0.01 | 2699 | 1861 | 2699:1861 | 7% | 0.61 |
| *Methylobacter* | 0.014 | 0.01 | 1810 | 1647 | 1810:1647 | 6% | 0.47 |
| *Methylacidiphilum* | 0.013 | 0.008 | 1880 | 1290 | 188:129 | 6% | 0.50 |


### 3.2.5 Functional gene composition

Of the total captured gene pool, 64% of sequence annotations were categorized by MG-RAST as coding for
metabolism (KO level 1). For metabolism pathways, the top three sub-categories (KO level 2) were distributed
across amino acid metabolism (32%), carbohydrate metabolism (27%) and energy metabolism (11%). Within the
energy metabolism category, $CH_4$ metabolism (PATH: KO00680) made up 17% of the captured genes (KO level
4) with a total of 109 genes coding for $CH_4$ metabolism.
The composition of the functional genes can be observed in the NMDS (fig 5). The NMDS displays the functional
genes grouped by HFM, MFM and LFM. Within the NMDS, we observe an overlapping between HFM, MFM
and LFM with a lack of distinct separation between clusters, indicating similar abundances and variation within




HFM, MFM and LFM. The PERMANOVA also calculated the coefficient of determination and revealed that only
7% ($R^2 = 0.07$, p ≥ 0.44) of the variation in functional genes can be explained by HFM, MFM and LFM. Finally,
we checked for differences between the means of HFM, MFM and LFM via pairwise distances and found no
significant difference (p ≥ 0.05).

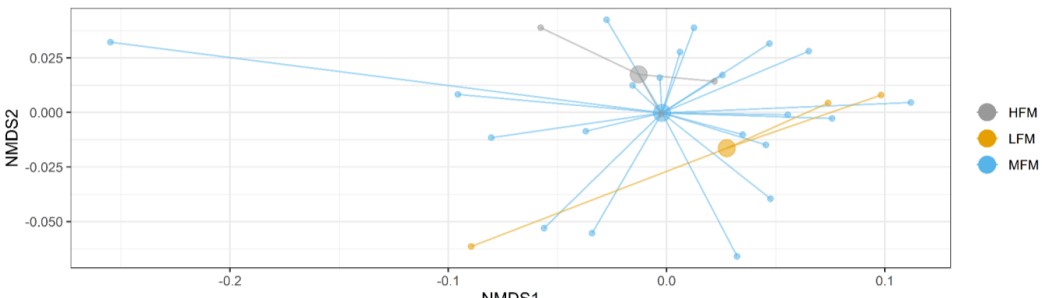


**Figure 5:** Nonmetric Multidimensional Scaling (NMDS) of functional genes using Bray-Curtis distances.
Samples analyzed at KO functional level 4 and colored by HFM, MFM and LFM (HFM n=3, MFM n=21,
LFM n=3).
In total, 21 genes of the 109 contributed to 70% of the cumulative sum (table 4, 5 and 6). When comparing HFM
to MFM and LFM, the Wilks' pairwise post hoc test revealed no significant difference between MFM (p ≥ 0.05)
and LFM (p ≥ 0.05). Within the two comparisons, we observed that heterodisulfide reductase subunit A (*hdrA*)
was the highest cumulative contributor to dissimilarity. In HFM, *hdrA* contributed to 13% of the cumulative total
(table 4). When comparing to LFM, *hdrA* contributed 10%, 3% lower than the HFH to MFM comparison (table
6). However, the permutation test revealed no significant difference between both MFM and LFM with regards
to abundance of *hdrA* (table 4). Within the top 70% of cumulative genes, only the abundance of particulate
methane monooxygenase (*pmoA)* was significantly higher in HFM when compared to MFM (p ≤ 0.01) (table 4).
As observed in the HFM comparisons, 21 genes contributed to the total 70% cumulative sum of all captured genes
when compared between HFM (table 4) and LFM (table 5). The Wilks' pairwise post hoc test revealed no
significant difference between HFM and LFM when compared to MFM (table 5). The largest contributor to the
cumulative sum of genes in MFM when compared to HFM and LFM was the *hdrA* gene. The *hdrA* gene
contributed to 13% in both comparisons. However, when comparing the abundances of the hdrA gene between
HFM and LFM, the decrease was non- significant. However, as identified in the earlier comparison, the abundance



of *pmoA* was significantly lower (p ≤ 0.01) in HFM when compared to MFM (table 4), but the same observation
was not observed when testing abundances in LFM (p ≥ 0.21) (table 5). Interestingly, *mcrA*, *mcrG* and *mtd* genes
were significantly higher in LFM when compared to MFM (p ≤ 0.02) (table 6).
As with the previous comparisons, 21 genes contributed to the cumulative sum of 70% within the LFM. However,
the order and amount contributed to dissimilarity were not the same. The Wilks pairwise post hoc test revealed no
significant difference between HFM and MFM when compared to LFM (p ≥ 0.05). Similarly, the *hdrA* gene was
the largest contributor to the cumulative sum in HFM and MFM (table 5 and 6). The largest difference in the
abundance of the *hdrA* gene occurred within HFM where the abundance increased by 3% (table 6). However, the
increase was identified as non-significant. Interestingly, the *mcrA* gene was highest in abundance in LFM (277),
54% higher than in HFM and 39% higher than MFM.





**Table 4:** Results of SIMPER analysis. Genes are ranked according to their average contribution to dissimilarity between MFM and HFM. Average abundances, ratio (between averages using the greatest common denominator), relative contribution of taxa and *p*-value of the permutation test (Probability of getting a larger or equal average contribution in random permutation of the group factor) are also included. A cut-off at a cumulative dissimilarity of 70% was applied.

| Contrast: MFM – HFM | Average | SD | Avg. MFM | Avg. HFM | Ratio | Relative Contribution (%) | p - value |
|---|---|---|---|---|---|---|---|
| *hdrA* - Heterodisulfide reductase subunit A | 0.040 | 0.044 | 429 | 225 | 143:75 | 13 | 0.72 |
| *cutL/coxL* - Carbon monoxide dehydrogenase large chain | 0.026 | 0.028 | 525 | 354 | 175:118 | 9 | 0.28 |
| *mcrA* - Methyl-coenzyme M reductase subunit A | 0.018 | 0.016 | 168 | 126 | 4:3 | 6 | 0.87 |
| *coxS* - carbon monoxide dehydrogenase small subunit S | 0.013 | 0.012 | 272 | 186 | 136:93 | 5 | 0.42 |
| *frhG* – coenzyme F420 hydrogenase gamma subunit G | 0.010 | 0.009 | 98 | 65 | 98:65 | 3 | 0.54 |
| *mtrA* – tetrahydromethanopterin S methyltransferase subunit A | 0.009 | 0.008 | 80 | 80 | 1:1 | 3 | 0.73 |
| *mvhA /vhuA /vhcA* -F420 non reducing hydrogenase subunit A | 0.008 | 0.008 | 86 | 45 | 86:45 | 3 | 0.66 |
| *cooS* – carbon monoxide dehydrogenase catalytic subunit S | 0.008 | 0.006 | 91 | 64 | 91:64 | 3 | 0.76 |
| *cutM/coxM* – carbon monoxide dehydrogenase medium subunit M | 0.007 | 0.007 | 145 | 99 | 145:99 | 3 | 0.42 |
| *fwdB/fmdB* – formylmethanofuran dehydrogenase subunit B | 0.007 | 0.007 | 83 | 59 | 83:59 | 2 | 0.95 |



| | | | | | | |
|---|---|---|---|---|---|---|
| *metF* – methylenetetrahydrofolate reductase NADPH | 0.007 | 0.006 | 128 | 81 | 128:81 | 3 | 0.27 |
| *pmoA* - particulate methane monooxygenase | 0.007 | 0.006 | 21 | 62 | 21:62 | 2 | **0.01** |
| *hoxH* - hydrogen dehydrogenase | 0.007 | 0.005 | 120 | 84 | 10:7 | 2 | 0.41 |
| *mttB* – trimethylamine corrinoid protein CO methyltransferase | 0.006 | 0.005 | 96 | 63 | 32:21 | 2 | 0.24 |
| *mtrH* - tetrahydromethanopterin S methyltransferase subunit | 0.005 | 0.004 | 59 | 42 | 59:42 | 2 | 0.92 |
| *fwdD/fmdD* – formylmethanofuran dehydrogenase subunit D | 0.004 | 0.004 | 38 | 33 | 38:33 | 2 | 0.76 |
| *mtrE* - tetrahydromethanopterin S methyltransferase subunit E | 0.004 | 0.004 | 43 | 32 | 43:32 | 1 | 0.86 |
| *hdrB* – heterodisulfide reductase subunit B | 0.004 | 0.003 | 55 | 37 | 55:37 | 2 | 0.91 |
| *mtrF* - tetrahydromethanopterin S methyltransferase subunit F | 0.004 | 0.003 | 38 | 25 | 38:25 | 1 | 0.95 |
| *mcrG* – methyl coenzyme M reductase gamma subunit | 0.004 | 0.003 | 44 | 52 | 11:13 | 2 | 0.79 |
| *CODH ACSA* – carbon monoxide dehydrogenase/acetyl CoA synthase subunit alpha | 0.004 | 0.004 | 42 | 20 | 21:10 | 1 | 0.65 |

411





**Table 5:** Results of SIMPER analysis. Genes are ranked according to their average contribution to dissimilarity between MFM and LFM. Average abundances, ratio (between averages using the greatest common denominator), relative contribution of taxa and *p*-value of the permutation test (Probability of getting a larger or equal average contribution in random permutation of the group factor) are also included. A cut-off at a cumulative dissimilarity of 70% was applied.

| Contrast: MFM – LFM | Average | SD | Avg. MFM | Avg. LFM | Ratio | Relative Contribution (%) | p – value |
|---|---|---|---|---|---|---|---|
| *hdrA* - Heterodisulfide reductase subunit A | 0.045 | 0.034 | 429 | 448 | 429:448 | 13 | 0.51 |
| *mcrA* - Methyl-coenzyme M reductase subunit A | 0.027 | 0.017 | 168 | 277 | 168:277 | 8 | 0.15 |
| *cutL/coxL* - Carbon monoxide dehydrogenase large chain | 0.021 | 0.016 | 525 | 573 | 175:191 | 6 | 0.82 |
| *fwdB/fmdB* – formylmethanofuran dehydrogenase subunit B | 0.014 | 0.009 | 83 | 161 | 83:161 | 4 | 0.05 |
| *mtrA* – tetrahydromethanopterin S methyltransferase subunit A | 0.014 | 0.008 | 80 | 143 | 80:143 | 4 | 0.10 |
| *coxS* - carbon monoxide dehydrogenase small subunit S | 0.011 | 0.007 | 272 | 311 | 272:311 | 3 | 0.87 |
| *mvhA/vhuA/vhcA* -F420 non reducing hydrogenase subunit A | 0.010 | 0.006 | 86 | 103 | 86:103 | 3 | 0.39 |
| *cooS* – carbon monoxide dehydrogenase catalytic subunit S | 0.010 | 0.007 | 91 | 125 | 91:125 | 3 | 0.36 |
| *frhG* – coenzyme F420 hydrogenase gamma subunit G | 0.009 | 0.008 | 98 | 113 | 98:113 | 3 | 0.70 |
| *mtrH* - tetrahydromethanopterin S methyltransferase subunit | 0.008 | 0.005 | 59 | 96 | 59:96 | 2 | 0.09 |
| *mtrE* - tetrahydromethanopterin S methyltransferase subunit E | 0.008 | 0.004 | 43 | 85 | 43:85 | 3 | 0.07 |





| | | | | | | |
|---|---|---|---|---|---|---|
| *mtrF* - tetrahydromethanopterin S methyltransferase subunit F | 0.007 | 0.004 | 38 | 77 | 38:77 | 2 | 0.05 |
| *fwdD/fmdD* – formylmethanofuran dehydrogenase subunit D | 0.007 | 0.004 | 38 | 76 | 1:2 | 2 | 0.06 |
| *mcrG* – methyl coenzyme M reductase gamma subunit | 0.007 | 0.004 | 44 | 84 | 11:21 | 2 | **0.02** |
| *hoxH* - hydrogen dehydrogenase | 0.007 | 0.005 | 120 | 153 | 40:51 | 2 | 0.44 |
| *mtd* – methylenetetrahydromethanopterin dehydrogenase | 0.006 | 0.004 | 38 | 76 | 1:2 | 2 | **0.02** |
| *cutM/coxM* – carbon monoxide dehydrogenase medium subunit | 0.006 | 0.005 | 145 | 148 | 145:148 | 2 | 0.85 |
| *frhB* – coenzyme F420 hydrogenase β- subunit | 0.006 | 0.004 | 49 | 80 | 49:80 | 2 | 0.08 |
| *metF* – methylenetetrahydrofolate reductase NADPH | 0.006 | 0.005 | 128 | 156 | 32:39 | 1 | 0.88 |
| *hdrB* – heterodisulfide reductase subunit B | 0.006 | 0.004 | 55 | 82 | 55:82 | 2 | 0.24 |
| *mcrB* – methyl coenzyme M reductase β- subunit | 0.005 | 0.003 | 32 | 63 | 32:63 | 2 | **0.02** |







**Table 6:** Results of SIMPER analysis. Genes are ranked according to their average contribution to
dissimilarity between HFM and LFM. Average abundances, ratio (between averages using the greatest
common denominator), relative contribution of taxa and p-value of the permutation test (Probability of
getting a larger or equal average contribution in random permutation of the group factor) are also
included. A cut-off at a cumulative dissimilarity of 70% was applied.

| Contrast: HFM – LFM | Average | SD | Avg. HFM | Avg. LFM | Ratio | Relative Contribution (%) | p – value |
|---|---|---|---|---|---|---|---|
| *hdrA* - Heterodisulfide reductase subunit A | 0.040 | 0.016 | 225 | 448 | 225:448 | 10 | 0.65 |
| *cutL/coxL* - Carbon monoxide dehydrogenase large chain | 0.032 | 0.016 | 354 | 573 | 118:191 | 8 | 0.28 |
| *mcrA* - Methyl-coenzyme M reductase subunit A | 0.028 | 0.012 | 126 | 277 | 126:277 | 7 | 0.25 |
| *coxS* - carbon monoxide dehydrogenase small subunit S | 0.017 | 0.005 | 186 | 311 | 186:311 | 5 | 0.20 |
| *fwdB/fmdB* – formylmethanofuran dehydrogenase subunit B | 0.016 | 0.007 | 59 | 161 | 59:161 | 4 | 0.06 |
| *mtrA* – tetrahydromethanopterin S methyltransferase subunit A | 0.015 | 0.006 | 80 | 143 | 80:143 | 3 | 0.13 |
| *metF* – methylenetetrahydrofolate reductase NADPH | 0.011 | 0.004 | 81 | 156 | 27:52 | 3 | 0.06 |
| *cooS* – carbon monoxide dehydrogenase catalytic subunit S | 0.010 | 0.006 | 64 | 125 | 64:125 | 3 | 0.37 |
| *mvhA/vhuA/vhcA* -F420 non reducing hydrogenase subunit A | 0.010 | 0.005 | 45 | 103 | 45:103 | 2 | 0.48 |
| *mtrH* - tetrahydromethanopterin S methyltransferase subunit | 0.009 | 0.004 | 42 | 96 | 7:16 | 3 | 0.09 |
| *frhG* – coenzyme F420 hydrogenase gamma subunit G | 0.009 | 0.004 | 65 | 113 | 65:113 | 2 | 0.59 |





| | | | | | | |
|---|---|---|---|---|---|---|
| *hoxH* - hydrogen dehydrogenase | 0.009 | 0.005 | 84 | 153 | 28:51 | 2 | 0.17 |
| *mttB* – trimethylamine corrinoid protein Co methyltransferase | 0.009 | 0.004 | 63 | 126 | 1:2 | 3 | 0.06 |
| *mtrF* - tetrahydromethanopterin S methyltransferase subunit F | 0.008 | 0.003 | 25 | 77 | 25:77 | 2 | 0.05 |
| *mtrE* - tetrahydromethanopterin S methyltransferase subunit E | 0.008 | 0.004 | 32 | 85 | 32:85 | 2 | 0.14 |
| *fwdB/fmdB* – formylmethanofuran dehydrogenase subunit B | 0.008 | 0.004 | 33 | 76 | 33:76 | 2 | 0.09 |
| *cutM/coxM* – carbon monoxide dehydrogenase medium subunit | 0.008 | 0.007 | 99 | 148 | 99:148 | 2 | 0.51 |
| *mcrG* – methyl coenzyme M reductase gamma subunit | 0.008 | 0.003 | 52 | 84 | 13:21 | 2 | 0.05 |
| *mtd* – methylenetetrahydromethanopterin dehydrogenase | 0.007 | 0.004 | 31 | 76 | 31:76 | 2 | 0.05 |
| *frhB* – coenzyme F420 hydrogenase β- subunit | 0.007 | 0.003 | 35 | 80 | 7:16 | 1 | 0.09 |
| *hdrB* – heterodisulfide reductase subunit B | 0.006 | 0.004 | 37 | 82 | 37:82 | 2 | 0.28 |






**4.0 Discussion:**
**4.1 Functional potential of the microbial community**
The dominant methane production pathway within our samples, as shown by the taxonomy and $\delta^{13}C$ signal of
emitted $CH_4$, was hydrogenotrophic methanogenesis. However, the presence of the genera acetoclastic
*Methanosaeta* and *Methanosarcina,* which possess a more diverse genome allowing them to perform
hydrogenotrophic, acetoclastic and methylotrophic methanogenesis, suggests that the community holds a
metabolic potential to produce $CH_4$ under altered environmental conditions. In addition, the presence of type *I*, *II*
and *Verrucomicrobia Proteobacteria* indicates that peatland methanotrophs hold the ability to oxidise $CH_4$ via
Ribulose monophosphate, Serine or Calvin-Benson-Bassham cycles. Therefore, if temperate peatland
environmental conditions which govern the production and consumption of $CH_4$ are to change under future climate
scenarios, we can expect $CH_4$ production and consumption to still occur, but possibly using alternative metabolic
pathways than currently observed.
The potential to produce and consume $CH_4$ under alternate environmental conditions is in agreement with other
metagenomics studies, which concluded that shifts from one dominant functional group to another can occur as
the microbial community already holds the metabolic potential to degrade soil organic carbon via different
metabolic pathways (Manoharan et al., 2017; Tveit et al., 2013). The rate of such shifts is dependent upon the
delivery of necessary products and environmental conditions conducive for methanogenesis. In the absence of
acetogenesis and fermentation, the less dominant functional groups (i.e. acetoclastic and methylotrophic
methanogens) may still remain dormant, due to the absence of necessary substrates to metabolize.
**4.2 Carbon flux characteristics**
We observed a high spatial variability in $CH_4$ flux, which is consistent with research conducted in other temperate
peatlands (Keane et al., 2021; Sun et al., 2013). The same pattern observed in $CH_4$ fluxes was also detected in
GPP and $R_{eco}$, but not in NEE. The high productivity, observed as high GPP, may be explained by a higher amount
of photosynthetic biomass within HFM, than in MFM and LFM. $R_{eco}$ followed the same pattern as $CH_4$ and GPP,
with highest observed flux in HFM and lowest in LFM. One potential reason for the high respiration from HFM
could be the significantly higher relative abundance of *pmoA*. The *pmoA* gene codes for the first step in
methanotrophy, where $CH_4$ is reduced to methanol, and finally $CO_2$, which is often used as a proxy for
methanotrophy (Franchini et al., 2015; Freitag et al., 2010). The higher abundance of *pmoA* may indicate a higher



rate of methanotrophy, which may help to explain the higher $CO_2$ flux respired by the methanotrophs in HFM. In
addition, higher plant productivity causes higher autotrophic respiration, which generally makes up ~50% of $R_{eco.}$
However, the vegetation may also be supplying more substrates to the microbial community, which in turn is
consumed and respired in the form of $CO_2$. Characteristics beyond our control, such as redox potential, oxic status
and substrate availability, may have additionally contributed to the variability in $CH_4$ and $CO_2$ fluxes (Bridgham
et al., 2013; Ström et al., 2012).
**4.3 The relationship between $CH_4$ magnitude and functional genes**
When comparing the dissimilarity of taxa and functional genes between flux categories, we discovered small
dissimilarities in taxonomy and functional genes. This result indicates that, while variation within carbon fluxes
is observed, the use of taxa and functional genes only explains a small amount of the variability and hence the
relationship is not statistically significant.
**4.3.1 Taxonomic**
We found that the microbial community had a higher diversity of methanogens than methanotrophs. This can be
the result of the high WTD limiting the habitable area of oxic-anoxic interface. The most abundant methanogen,
*Methanoregula*, has been frequently detected in ombrotrophic peatland ecosystems and appears to dominate in
sites with high *Eriophorum spp.* (Andersen et al., 2013, Chroňáková et al., 2019, Lin et al., 2014, Preston et al.,
2012). The tussock building *E. vaginatum* provides a habitable environment for fermenters and syntrophic bacteria
where substrates such as $H_2$ and $CO_2$ for hydrogenotrophic methanogenesis are most likely more available due to
the increase in oxygen provided to the peat through aerenchyma tissue of the plant (Chroňáková et al., 2019;
Preston et al., 2012).
When comparing our results to other metagenomic approaches, we find a higher diversity of methanogens than
previous research. We identified 20 genera of methanogens, while Lin et al. (2012) detected 16 genera of
methanogens using a whole metagenomic approach. We observe slightly higher diversity than studies in other
ombrotrophic peatland environments, where sequences belonging to the orders *Methanomicrobiales*,
*Methanobacteriales*, and *Methanosarcinales* were detected (Galand et al., 2003; Horn et al., 2003b; He et al.,
2015). However, it is difficult to conclude whether our results differ from other studies because of biological
factors, different site characteristics or the addition of newly sequenced genomes within the databases used
between studies.



The composition of the microbial community was dominated by hydrogenotrophic methanogens. The dominant
genus, *Methanoregula*, is recognized as an indicative genus to ombrotrophic peatlands (Andersen et al., 2013;
Chroňáková et al., 2019; Lin et al., 2014; Preston et al., 2012), and this is further confirmed by our results. This
result was expected as methanogenic communities in ombrotrophic bogs differ significantly compared to fen
ecosystems (Horn et al., 2003b). However, the presence of acetoclastic and methylotrophic methanogens within
our samples indicates a high functional potential of ombrotrophic bogs with possibilities to switch between
dominant methanogenic functional groups. Theoretically, if conditions were to shift within the peatland to favor
acetoclastic or methylotrophic methanogenesis, the microbial community holds the functional potential to
continue producing $CH_4$ with little to no delay in transition period. This conclusion is of course made assuming
that the necessary substrates and environmental conditions are met.
When comparing the abundances of methanotrophs between HFM, MFM and LFM, we identified that the top 3
contributors to the cumulative sums, *Methylocella*, *Methylosinus* and *Methylobacter*, did not change significantly
in abundance or order of highest contributor. We expected that a higher proportion of type II and *Verrucomicrobia*
methanotrophs would contribute higher to the cumulative sums due to their ability to resist acidic conditions found
in bog environments (Hakobyan and Liesack, 2020; Dedysh, 2002, 2009), and this was confirmed by our results.
Both type II *Methylocella* and *Methylosinus* are well adapted to the cold and acidic conditions common in northern
ombrotrophic peatlands. These physiological traits explain why type II *Alphaproteobacteria* were dominant over
type I *Gammaproteobacteria* and this is consistent with other research conducted in ombrotrophic bogs
(Hakobyan and Liesack, 2020; Chen et al., 2008; Dedysh, 2002, 2009). However, the presence of thermophilic
and halophilic *Verrucomicrobia* and *Gammaproteobacteria* methanotrophs, while lower in abundance, were also
detected in each category. This indicates a tolerance to the acid and cold conditions experienced within northern
ombrotrophic peatlands. These results, similar to those observed in the methanogen community, indicate that the
methanotroph community hold the ability to continue to oxidise $CH_4$ under alternate environmental conditions.

### 4.3.2 Functional genes

The functional gene composition of methanogens and methanotrophs does not hold a strong relationship to the
magnitude of $CH_4$ flux, contrary to results found by Zhang et al. (2019) were the authors observed significant
correlation between mcrA and $CH_4$ flux. However, Zhang et al. (2019) only targeted *mcrA* and *pmoA* when
analysing their results, while our approach used a wider diversity of methanogenesis and methanotrophy related
genes, which may have contributed to the observed difference. In our comparison, we observed small variations


in the relative abundance of genes when compared between HFM, MFM and LFM. The NMDS analysis agreed
with the PERMANOVA and displayed overlap between samples with a distinct lack of cluster separation. This
result indicates that the composition and relative abundance of functional genes has little variation between HFM,
MFM and LFM.
The top three genes that contributed the most to the dissimilarities between HFM, MFM and LFM were *mcrA,*
*hdrA* and *coxL.* Both *mcrA* and *hdrA* genes act as key enzymes in the biological formation of $CH_4$ and these genes
are shared across hydrogenotrophic, acetoclastic and methylotrophic methanogens. The *mcrA* catalyzes the
conversion of methyl-coenzyme M and coenzyme B into $CH_4$ and the heterodisulfide of coenzyme M (HS-CoM)
and coenzyme B (HS-CoB) (Scheller et al., 2010; Thauer, 2019). Subsequently, CoM and CoB must be reduced
to regenerate the CoM-SH and CoB-SH thiols which are used as electron donors by *mcrA,* which is then catalyzed
by *hdrA* (Scheller et al., 2010; Buan et al., 2011). Therefore, a co-dependence between *mcrA* and *hdrA* exists and
this is essential for the biological formation of $CH_4$. Due to the close nature of the two genes, targeting transcripts
of *hdrA* may be important in future research.
We assumed that the abundance of *mcrA* genes would be higher in HFM when compared to MFM and LFM in
accordance with previous research (Franchini et al., 2015; Liebner et al., 2012). However, the opposite was
discovered with the average relative abundances of *mcrA* lower in HFM when compared to MFM and LFM. This
result is surprising, as previous research has found a significant relationship between key genes such as the *mcrA*
and the magnitude of $CH_4$ flux (Freitag et al., 2010; Zhang et al., 2019). We believe that the analysis of *mcrA*
transcripts, rather than gene abundance, would yield a stronger relationship to the $CH_4$ flux. While a close
relationship of *mcrA* gene abundance to transcripts was observed by Franchini et al. (2015), gene abundance may
not be the most effective in explaining small differences between flux categories. Rather, the use of gene
transcripts may be a more appropriate method (Franchini et al., 2015; Freitag et al., 2010).
In addition to *mcrA* and *hdrA,* the presence of carbon monoxide (CO) dehydrogenase (*cooS*, *coxL*, *coxM*, *coxS*,
*cutL, cutM*) was of particular interest. These genes code for CO dehydrogenase and are involved in the Acetyl-
CoA pathway, which is not directly included in methanogenesis. Comparatively little is known today of the
ecology, physiology, and biochemistry of CO utilization by methanogens (Ferry, 2010; Fischer et al., 1931). Only
a few species are reported to metabolize CO, including *Methanosarcina*, which contributed 5% to 6% of the
cumulative sums within our comparisons. However, according to Ferry (2010) it is not yet known if CO is a viable
energy source for methanogens in peatland environments. Furthermore, the presence of six genes that code for



CO dehydrogenase within the top 70% of cumulative sums indicates that if CO is a viable substrate, a high
functional potential could exist within peatland environments to use this lesser known substrate during
methanogenesis.
It is important to note that carbon monoxide dehydrogenase and *hdrA* genes are not strictly utilized by
methanogens. A wide variety of microbes, including *Acetogens,* sulfur oxidizing *Archaea* and *Bacteria,* utilize
the above-mentioned genes (Ernst et al., 2021; Ferry, 2010; Maupin-Furlow and Ferry, 1996). Therefore, the
distribution of how many genes are strictly related to methanogenesis can be difficult to determine.

**4.4 The relationship between microbial diversity and the magnitude of $CH_4$ flux**

The Shannon α-diversity of 2.38 indicates a low diversity. An et al. (2019), found that peatland environments hold
an average Shannon α-diversity index of 6.8. However, the lower diversity observed in our is most likely due to
the targeted approach, which only enriches taxa related to $CH_4$ metabolism. The targeted approach, combined
with the filtering of taxa that exclude other microbial groups, which if included within the analysis, would better
represent peatland environments.
Low dissimilarity was observed between the mesocosms when calculating the $\beta$- diversity. HFM held the highest
dissimilarity indicating that as the $CH_4$ flux increases, the abundance and variability of microbe's increase. As the
magnitude of $CH_4$ flux reduced, the abundance and variability of methanogens and methanotrophs decreased. This
trend indicates that $\beta$- diversity may act as a proxy for $CH_4$ emissions, contrary to results found by Zhang et al.
(2019) who concluded that abundance, rather than composition mainly affects $CH_4$ emissions. However, due to
the low replication in HFM and LFM, further research is needed to make this conclusion.

**4.5 $\delta^{13}C$ of emitted $CH_4$ and proportion of taxa**

The $\delta^{13}C$ analysis and presence of multiple hydrogenotrophic methanogens indicated that the dominant metabolic
pathway observed within the mesocosms was hydrogenotrophic methanogenesis. All flux categories returned the
$\delta^{13}C$ signal within the hydrogenotrophic range ($\delta^{13}C = -110‰$ to $-60‰$) (Chanton, 2005; Whiticar, 1999). This
is not a surprising result, as there appears to be a pattern in northern and temperate wetlands of increasing
hydrogenotrophic methanogenesis going from minerotrophic peats to ombrotrophic acidic bogs, similar to
conditions observed with our site (Galand et al., 2010; Holmes et al., 2015; Horn et al., 2003a). Furthermore, the





positive correlation between $\delta^{13}$C-CH$_4$ to CH$_4$ emission rate indicates the CH$_4$ emission to be mostly controlled
by the trophic status for methanogenesis, rather than methanotrophy (Hornibrook, 2009).
**5.0 Conclusion**
In this paper, we addressed differences in the composition of taxonomy and functional genes of CH$_4$ producing
and consuming microbes between three flux categories: HFM, MFM and LFM. In addition, we determined that
$\beta$-diversity increases in HFM when compared to the MFM and LFM categories, and we observed that the $\delta^{13}$C of
emitted CH$_4$ matches the dominant taxonomic functional group.
We observed small differences in the composition of both taxa and functional genes between flux categories. This
indicates that, although we observe high spatial variability in CH$_4$ fluxes, we cannot explain this variability by
taxonomic composition and functional genes alone. Interestingly, we observed that $\beta$-diversity was higher in HFM
when compared to MFM and LFM – indicating that diversity may be a plausible proxy for CH$_4$ fluxes.
The dominant methanogen, *Methanoregula,* made up over 50% of the community composition. This, in
combination with the remaining hydrogenotrophic methanogens observed within the community composition,
matched the observed $\delta^{13}$C isotopic signature of emitted CH$_4$. This indicates that the dominant metabolic pathway
in the Fäjemyr peatland is hydrogenotrophic methanogenesis. However, the presence of acetoclastic and
methylotrophic taxa plus type I, II and *Verrucomicrobia* methanotrophs indicates that the microbial community
holds the ability to produce and consume CH$_4$ via alternate metabolic pathways. This is important in terms of
future climate scenarios where peatlands can expect altered nutrient status, hydrology or peat chemistry. If this
happens, we can expect that there will be methanogen and methanotrophs present to continue to produce and
consume CH$_4$ due to the potential for alternate metabolic pathways.
Our results show that genetic potential is of minor importance in explaining small scale variability of CH$_4$ fluxes
observed in peatland environments. Additional proxies to understand this variability may be found in gene
expression studies where activity levels are better represented rather than genetic potential. With the modeling
community working continuously to build robust predictions of peatland CH$_4$ emissions (Chadburn et al., 2020),
the need for inclusion of genomic data may be considered. With this knowledge, the combination of traditional
CH$_4$ drivers, metagenomics and metatranscriptomic studies could increase our understanding of how and at what
rate the key CH$_4$ producing and consuming microorganisms' function in peatland ecosystems. This information



on microbial diversity is necessary on both temporal and spatial scales for the development of more robust models
to accurately predict upcoming emissions under future climate scenarios.

**Data availability**

The annotated metagenomes are available at the MG-RAST server under the project ID: 91052. In addition, all
raw sequences will be made public via NCBI Bio Project ID: PRJNA691743 upon acceptance of this manuscript.

**Code availability**

Code used in the analysis can be found at https://github.com/joel332/Analysis-of-captured-metagenomic-
data/tree/main.

**Author contributions**

JW, LS and DA planned the experiment; JW, LS, JR performed the measurements; JW and VL analyzed the data,
JW wrote the draft; JW, LS, JR, VL an DA reviewed and edited the manuscript.

**Competing interests**

The authors declare that they have no conflict of interest.

**Acknowledgments**

We would like to thank Pia and the staff from the Centre for Genomic Research, Ulrika from Roche Diagnostics
Scandinavia, Frans-Jan Parmentier, colleagues, friends and family for the valuable discussions and Oskar Ström
for laboratory assistance. The data handling was enabled by resources in project (SNIC 2019/8-365) provided by
the Swedish National Infrastructure for Computing (SNIC) at UPPMAX, partially funded by the Swedish
Research Council through grant agreement no. 2018-05973.

**Financial support**

We would like to thank the Swedish research council for environment, agricultural sciences and spatial planning
(FORMAS - 997-610) and the Crafoord Institute (20200738) for financial support.

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
