# Peer review of "Genetic functional potential displays minor importance in explaining spatial variability of methane"

_Biogeosciences, 2021_

## Author Comment (AC1)

**RC1**: 'Comment on bg-2021-353', Anonymous Referee #1, 09 Feb 2022

General comments:

White et al. collected soil samples and brought them to the laboratory for an observational mesocosm study under controlled conditions. They measured a suite of ecosystem processes including net ecosystem exchange, respiration, $CH_4$ flux, and stable isotope analysis. They performed captured metagenomics to examine the microbial community membership and gene content, including organisms directly implicated in $CH_4$-cycling processes. The paper does not in its current form seem to be driven by any particular hypothesis, but rather is focused on examining fluxes and microbial communities under laboratory conditions that approximate field conditions.

It seems that much of the focus of the paper relies on distinguishing the samples into 3 categories (HFM, MFM, and LFM), with the first and last category only coming from single samples (with 3 technical replicates each). In my opinion this makes the paper more about how two outliers differ from the rest of the samples than about the relationship between fluxes and communities in general. If distinguishing samples among three tiers is how you want to proceed, why not rank all samples by their fluxes and then divide them evenly into these three categories? In its current form this categorization seems to make your statistics very unbalanced.

In terms of the level of inference the authors make, there are several instances that I found problematic. For instance, the authors claim that HFM has higher B diversity than the rest, yet this relationship was not significant, and was also based on a mis-balanced design. The authors also state several times that just because they see several types of methanogens/methanotrophs that these communities ought to continue functioning under future climate scenarios. Without performing and experimental test of this hypothesis these types of speculation should not be in the paper, and especially not a main takeaway (e.g. in the conclusions).

*Authors response: We would like to thank the reviewer for the comments supplied and we have made changes accordingly. With regards to the comment on how we distinguished categories we understand your concerns. The establishment of high, medium and low emitting mesocosms was a result we discovered after the measurement period and initially we did not have that study aim. However, after discovering that two mesocosms were acting significantly different from the others we decided to peruse whether the genomic data could help explain this variability. Rather than ranking all samples by their fluxes and then divide them evenly into these three categories we performed a randomization test to see whether these mesocosms could be separated and found that only mesocosm 4 and mesocosm 9 were giving statistically higher (mesocosm 4) and lower (mesocosm 9) fluxes. In addition, the statistical tests which we have performed have taken into account the uneven study design and low replication.*

*With regards to our conclusions made from the beta diversity section we have made considerable changes to account for your concerns. Rather than using such a definitive conclusion from our results we have highlighted that we observe a trend but further research is necessary to confirm what we propose. Our conclusions on how communities ought to continue under future climate scenarios has been made from reading existing literature. The literature states the environmental*

*conditions to what the microbes find tolerable. We have reflected this in our updated text in the comments below.*

*Thank you once again for your constructive criticism of our work and please don't hesitate to contact us with further questions if needed. Please see detailed responses to specific comments below.*

Specific comments:

Line 45: add 'the': is the second most. Also important seems like it needs a qualifier: important for climate?

*Authors response: This alteration has been made according to reviewer's suggestion*

70-75: nice summary of controls.

*Thank you. You don't often receive compliments in the peer review process.*

86-88: "The targeting of…", this sentence may not be necessary for the scope of your paper. Just a suggestion.

*Authors response: This sentence has been removed according to the reviewer's suggestion*

105-108: Just as a comment, this reads a bit like an advertisement.

*Authors response: Thank you for the comment. We have chosen to keep the original line as it emphasizes the reason why we chose to use the captured metagenomics approach opposed to 16s and whole metagenomics.*

139-140: Do you mean to say that placement of the mesocosms was varied bi-weekly? "Rotating" could be interpreted as simply turning them.

*Authors response: Thanks for the observation, we have added the following for clarification:*

*Original: The mesocosms were rotated bi-weekly to minimize the effect of spatial variations in growth conditions.*

*Edited: In response, the mesocosms were rotated to different positions on the table bi-weekly to minimize the effect of spatial variations in growth conditions.*

147: Perhaps change section header to "Flux measurements of mesocosms" for clarity.

*Authors response: This alteration has been made according to the reviewer's suggestion*

171: I assume you mean 'stored at -20C', not '20C'. Please clarify.

*Authors response: Yes, you are correct. An typo on our behalf. The change has been made accordingly*

260: It becomes difficult to follow the text when there are so many abbreviations. Perhaps consider not abbreviating.

*Authors response: This alteration has been made according to the reviewer's suggestion. HFM will become mesocosm 4, MFM will become medium emitting mesocosms and LFM will become mesocosm 9.*

259-263: in a similar vein, there are a lot of abbreviations in this section that haven't been defined yet in the results section. Consider naming them here (or using the full words) for clarity.

*Authors response: This alteration has been made according to reviewer's suggestion. See above comment.*

302-303: you don't have to italicize the word phylum.

*Authors response: Yes, you are correct. The alteration has been made accordingly*

303-304: how do you know that this is due to environmental conditions? This sounds presumptuous without an explanation.

*Authors response: the literature generally states that NC10 methanotrophs are mesophilic and neutrophilic, therefore we did not expect to find such methanotrophs in an acidic bog. We have added the following to justify the statement:*

*Original: Due to the environmental conditions no methanotrophs from the NC10 Phylum were detected*

*Edited: No NC10 methanotrophs were detected, as expected, since this functional group is generally only reported in mesophilic and neutrophilic conditions while the conditions at the Fäjemyr mire are acidic and cold.*

315: bacteria can be lower case and not italicized here.

*Yes, you are correct. The alteration has been made accordingly*

321-329: I do not like that you can comparing Beta diversity among groups that have very uneven sample numbers. Remind the readers in this section how many samples are in each group.

*Authors response: We have included the number of replicates in both methodology and also in the figure text of figure 4. For additional clarity we have added the number of replicates for each group within the text. In addition, we believe that renaming HFM to "mesocosm 4" and LFM to "mesocosm 9" will add further clarity to the reader. Furthermore, rather than plotting using the*

*boxplot we have change figure 4 to a NMDS plot. Here the reader can see the number of replicates clearer and interpret the results for Beta diversity more easily.*

*Original: β-diversity, which measures the change in diversity of species from one category to another, was measured as mean distance to the group centroid and highest in HFM (fig 4). HFM resulted in an average distance to median of 0.046, followed by MFM (0.042) and LFM (0.031).*

*Edited: β-diversity, which measures the change in diversity of genera from one category to another, was measured using dissimilarity indices. Furthermore, the dissimilarity indices were calculated using the average distance of group members to the group centroid which is shown in fig 4. In fig 4, we observe an overlap between mesocosm 4, medium emitting mesocosms and mesocosm 9, and a lack of distinct separation between clusters, indicating similar diversity and variation within all mesocosms. We observed the largest dissimilarity between mesocosm 4 (n = 3) to medium emitting mesocosms (n = 21) with a mean difference in dissimilarity of 0.023. The lowest difference in dissimilarity was observed between the medium emitting mesocosms and mesocosm 9 (n = 3) with a 0.005 difference in dissimilarity between groups. Due to a high variation and lack of spatial replications for mesocosm 4 and 9, this relationship was observed as non-significant. As the taxonomic data is a subset (i.e. only including methanogen / methanotroph taxa) of all the taxonomic sequences contributing to the whole metagenomic community, values for β-diversity are low. However, the differences between centroids indicates that communities of methanogens and methanotrophs become more similar to each other as the magnitude of flux decreases.*

327-329: is this the Beta diversity of the whole community or just a subset of methanogens and methanotrophs? This sentence would lead me to believe it is the latter and if that is the case this sound be clarified in the section header as well as the text.

*Authors response: The values for beta diversity shown in the paper are analysed of the subset of "captured" taxa. The taxa were filtered to only include methanogens and methanotrophs by removing off target taxa using the MG-RAST filter. This is written in the methods section 2.6 but may not be clear enough. We have made the alteration to the text to compensate for this.*

*Original: Although the values for β-diversity are low, the differences between centroids indicates that communities of methanogens and methanotrophs become more similar to each other as the magnitude of flux decreases.*

*Edited: As the taxonomic data is a subset (i.e. only including methanogen / methanotroph taxa) of all the taxonomic sequences contributing to the whole metagenomic community, values for β-diversity are low. However, the differences between centroids indicates that communities of methanogens and methanotrophs become more similar to each other as the magnitude of flux decreases.*

Fig 4: show points on the same boxplot graph so readers can understand visually that you are not comparing equal sample numbers.

*Authors response: We thank you for this observation. Upon reflection we decided that beta diversity is better displayed using NMDS rather than boxplots. With this alteration, the reader can clearly see the number of replicates and the variation between the groups. In addition, we have added clarification to the number of replicates in the text and figure text as stated above.*

340: It is not clear in the text why you are doing this analysis three times and reporting three tables. Perhaps you could choose the one most important to your narrative and put the other two in the supplementary? The three tables have identical table legends so it really is not obvious what is distinguishing them and what the reader should take away.

*Authors response: Table 1, 2 and 3 are the comparisons between each flux category. Within each table text and table headings the reader can observe which groups are being compared i.e. "Taxa are ranked according to their average contribution to dissimilarity between medium emitting mesocosms and mesocosm 4". This must not be as clear as we originally thought. In response, we have added a clearer table title to the table text.*

Same comment for Tables 4, 5, and 6.

*Please see comment above*

433-436: We do not know anything about the environmental tolerances of these organisms. I think it is too speculative to make any inferences about the future prospects of these processes under climate change scenarios based on the sole observation that there are members of these different groups present.

*Authors response: Our conclusions made on how communities may continue under future climate scenarios and have been made according to existing literature. The literature states taxa have the ability to exist within certain environmental boundaries.*

*Edited: (Discussion) Theoretically, if conditions were to shift within the peatland to favor acetoclastic or methylotrophic methanogenesis, the microbial community already holds the functional potential and specific environmental tolerances to continue producing $CH_4$.*

*(Discussion) This indicates a tolerance to the acid and cold conditions experienced within northern ombrotrophic peatlands. These results, similar to those observed in the methanogen community, indicate that the methanotroph community holds the ability to continue oxidizing $CH_4$ under alternate environmental conditions.*

*(Conclusion) This is important in terms of peatlands under future climate pressure where we may see altered nutrient status, hydrology or peat chemistry. If this shift in peatland status happens, our results indicate that the $CH_4$ producing and consuming microbial community hold the potential to be dominated by alternate functional groups (i.e. acetoclastic) than what we observe now, thus*

*holding the functional potential to continue the production and consumption of CH₄ at Fäjemyr mire under the correct environmental conditions.*

489: "with little to no delay in transition period" – what are you basing this statement on?

*We have removed this statement for clarity of the reader.*

569-570, 574-575: but these differences were not statistically significant. This should not be in your conclusions.

*Authors response: The statements regarding beta diversity has been removed from conclusion but remain in the discussion. After some thought, we agreed that even though the results are interesting the trend is not significant therefore we should not be included within the final conclusions.*

581-584: If your study experimentally manipulated the environment of these mesocosms to examine future climate scenarios then you might have the data to back up this sentence. I think that just because you are seeing representatives of these different groups does not tell us anything about the future prospects of these microbes or the processes they perform.

*We have revised this sentence to include less definitive wording, please see alterations below.*

*Original: This is important in terms of future climate scenarios where can expect altered nutrient status, hydrology or peat chemistry. If this happens, we can expect that there will be methanogen and methanotrophs present to continue to produce and consume CH₄ due to the potential for alternate metabolic pathways.*

*Edited: This is important in terms of peatlands under future climate pressure where we may see altered nutrient status, hydrology or peat chemistry. If this shift in peatland status happens, our results indicate that the CH₄ producing and consuming microbial community hold the potential to be dominated by alternate functional groups (i.e. acetoclastic) than what we observe now, thus the production and consumption of CH₄ may continue at Fäjemyr mire under the correct environmental conditions.*

---

## Author Comment (AC2)

*Comment on bg-2021-353*
*Anonymous Referee #2*
*Referee comment on "Genetic functional potential displays minor importance in explaining spatial variability of methane fluxes within a Eriophorum vaginatum dominated Swedish peatland" by Joel Dawson White et al., Biogeosciences*

*Discuss.,*
*https://doi.org/10.5194/bg-2021-353-RC2, 2022*

*Factors that affect and/or govern methane emission from wetlands are of great interest because better understanding of the influential factors would enhance the predictability of methane emission from wetlands when subjected to environmental changes. This paper aims to assess the functional potential impact of CH4 producing and consuming microbes on the magnitude of CH4 flux. The authors concluded that the functional potential [of the methane cycling community] plays a minor role in explaining the observed differences in methane flux categories (HFM, MFM, LFM).*

*The authors thank the reviewer for their comments. We have addressed the major issues in individual sections rather than writing a lengthy text at the bottom. Please find our responses below. In addition, each specific comment has been addressed individually.*

*Major issues:*
*The key weakness of this paper is the use of genetic information of the methane cycling community alone in an attempt to address the scientific question the author set out*

*Authors response: We agree that the $CH_4$ cycle is not simple and does not operate individual of other processes. However, to reduce the complexity of this system, we kept temperature and light levels the same among treatments while the water table was kept stable throughout the experiment (section 2.2). This minimizes the influence of these drivers on spatial variability, which should enhance differences arising from $CH_4$ cycling functional genes and taxonomy. We believe that using our targeted approach is a strength, rather than a weakness as we can observe many functional genes used across multiple metabolic pathways, which cannot be achieved when using 16s studies. In addition, we were able to make conclusions not observed in more complex whole metagenomic studies.*

*Firstly, the text does not provide clearly the reasoning of why the authors hypothesized that the differences in the measured methane flux (categories) could be explained by shifts in the composition of the methane cycling taxa in the 9 mesocosms. It would be to lay out the logic.*

Authors response: We aim to clarify this by adding additional text to the introductory section. See revisions below.

*Original: CH₄ emissions from natural wetlands are known to exhibit both spatial and temporal variability (Crill et al., 1988; Sun et al., 2013). The spatial variability makes wetland CH₄ emissions difficult to model and predict (Wania et al., 2009, 2010), as CH₄ emission within similar environmental conditions (i.e. ecotype) can vary by several orders of magnitude without an apparent explanation (Bridgham et al., 2013). According to current knowledge, both production and consumption of CH₄ within peatland ecotypes is driven by (i) water table depth (WTD), which determines the thickness of oxic and anoxic zones; (ii) plant species composition, which provides substrates and plant mediated transport of CH₄ to the atmosphere; (iii) soil temperature, which affects the rate of microbiological processes; and (iv) substrate availability for biogeochemical processes such as methanogenesis and methanotrophy (Joabsson et al., 1999; Korrensalo et al., 2018; Mastepanov et al., 2013; Strack et al., 2004; Ström et al., 2015).*

*Authors revision L66-75: CH₄ emissions from natural wetlands are known to exhibit both spatial and temporal variability (Crill et al., 1988; Sun et al., 2013). The spatial variability makes wetland CH₄ emissions difficult to model and predict (Wania et al., 2009, 2010), as CH₄ emissions under similar environmental conditions (i.e. ecotype) can vary by several orders of magnitude without an apparent explanation (Bridgham et al., 2013). According to current knowledge, the magnitude of CH₄ fluxes in peatlands is driven by (i) water table depth (WTD), which determines the thickness of oxic and anoxic zones; (ii) plant species composition, which provides substrates and plant mediated transport of CH₄ to the atmosphere; (iii) soil temperature, which affects the rate of microbiological processes; and (iv) substrate availability for biogeochemical processes such as methanogenesis and methanotrophy (Joabsson et al., 1999; Korrensalo et al., 2018; Mastepanov et al., 2013; Strack et al., 2004; Ström et al., 2015). Interestingly, the microorganisms which produce and consume CH₄ are either not included in models or assume that no spatial variability occurs in the functional potential of these communities (Chadburn et al., 2020). Rather, they picture the below ground microbial community as a uniform black box. Therefore, the need to research whether the functional potential of the microbial community contributes to the spatial variability has become more important in improving model predictions of CH₄ emissions from peatlands.*

**And the data presented in this manuscript indicated that although 9 mesocosms contain different number of tillers (Fig. 2), they exhibited statistically comparable magnitude of CH4 fluxes. Authors also pointed out their understanding that gene expression would be a better proxy.**

*Authors response: The reviewer is correct in saying that the data indicated that 9 mesocosms contains different number of tillers, a common driver of CH₄ emissions. However, the 9 mesocosms did not exhibit statistically comparable magnitudes of CH₄ fluxes, rather significantly higher or lower CH4 flux depending upon the group (section 3.1.1). This difference*

*can be observed in figure 1 and was the reason for establishing three flux categories (LFM, MFM and HFM), and the basis for the paper. The mention of gene expression on line 587 is merely a suggestion for future research used in our conclusion.*

*Secondly, only abundance data of these methane cycling taxa relative to each other (i.e. the methane cycling community) was stated (or available). It is uncertain to this reviewer that how the PCR steps in the "captured metagenomics" analysis might have altered such relative abundance. And the use of "captured metagenomics" has, to the disadvantage of the study, prevented one from knowing the abundance of the methane cycling community relative to the total microbial community, as such relative abundance would be helpful to hint the proportion of the whole methane cycling community. These may explain why this study does not find significant correlation between the so-called "functional gene abundance" with the observed CH4 flux categories, when compared to Zhang et al. (2019, cited in this manuscript) that showed a positive correlation of gene abundance (absolute mcrA gene copy number) and methane flux. The use of relative abundance to the specific functional group is less robust when compared to absolute gene copy numbers. Additionally, it was not obvious that the calculation of the "abundance data" was explained in details or with clarity. It will be helpful for the reads to understand how the sequencing data was process to obtain the abundance. Some of the above mentioned points could be addressed by writing, but sadly, this weakness is a fundamental flaw that transcends through this manuscript, and affects the robustness of the analysis and interpretation*

*Authors response: The research has been conducted using the captured metagenomics approach. We used this approach as we wanted to narrow our research question and try not to overcomplicate our conclusions by including a broader whole metagenomic approach. As this approach uses custom designed probes to target sequences of interest, any off-target sequences (i.e. non methanogen/methanotroph) within our dataset must be ignored and we cannot trust those values to be correct. Therefore, we chose to exclude any other taxa other than methanogens and methanotrophs. During the PCR step, 7 cycles were used for libraries with a genomic DNA input of 150 ng, and 5 cycles where the input was 1 μg to minimise any risk of PCR biases (section 2.5.3). In addition, we did not see any correlation between the samples with low amount of DNA versus those with high. We use the term relative abundance throughout this paper as we cannot call it absolute abundance since that is not what we are measuring; Rather, when using next generation sequencing, we always get relative abundances.*

*Our results are different from Zhang et al. because we have used a different approach. Zhang et al used absolute abundance, while we focused on the methane producing / consuming community. The main aim of our research was to address whether the composition of both $CH_4$ producing and consuming taxa/functional genes shift in dissimilarity in response to variations in $CH_4$ fluxes, and not whether individual genes such as mcrA correlate to the magnitude of $CH_4$ flux. This is already well established by Zhang et al. and other studies. Sequence data and calculated abundance were all obtained through the MG-RAST annotation pipeline which is*

*referenced in section 2.6. We choose not to include all the steps within the MG-RAST pipeline since the pipeline is a well-established method and readers can access more details through the Meyer et al., 2008 reference.*

*Specific comments:*

*L 240-241, what data was being transformed? Was standardization or normalization done on the post-QC data?*

*Authors response: We used a double root transformation on abundance of taxa and functional genes as a form of normalization. We clarified this in the text as follows:*

*Original: Input data for the PERMANOVA was double root transformed to reduce the influence of highly abundant taxa and genes.*

*Authors revision: Taxonomic and gene abundances data for the PERMANOVA was double root transformed to reduce the influence of highly abundant taxa and genes.*

*L45 Missing "the" before "second most important", and please delete "has" in "has in the atmosphere"*

 *Authors response: This change has been made according to the reviewer's suggestion*

*L89 Is mmoX a commonly targeted gene in CH4 research? mmoX gene codes for the soluble methane monooxygenase, which is known to use substrates other than CH4. Did the authors mean to say mmoX or particular methane monooxygenase (pmoA)?*

*Authors response: As stated in the text mmoX is often targeted in similar experiments, however as we particularly focus upon pmoA in this manuscript and believe the swap to pmoA to be more appropriate. This change has been made according to the reviewer's suggestion*

*Original: In $CH_4$ research, key genes such as methyl coenzyme M reductase (mcrA) and methane monooxygenase component A alpha chain (mmoX) are often targeted to determine community composition and functional potential*

*Authors revision: In $CH_4$ research, key genes such as methyl coenzyme M reductase (mcrA) and particulate methane monooxygenase subunit A (pmoA) are often targeted to determine community composition and functional potential*

*L98 "are detected" should be "to be detected"*

Authors response: This change has been made according to the reviewer's suggestion

*L100 (there may be a better place to mention the following) mcrA gene is for detecting both methanogenic and methanotrophic archaea. Anaerobic methanotrophs (ANMEs) have been detected, albeit at very low abundance, in wetlands and permafrost-affected areas. Nonetheless, ANMEs have not been mentioned in this manuscript. In this study, Methanosarcinales were found among the methanogens, and Methanosarcinales contains ANMEs. Authors are suggested to investigate further whether ANMEs have contributed to the taxonomic and functional diversity in their data.*

*Authors response: we have added a shot passage of text within the discussion to reflect this. Members of the order Methanosarcinales were included in the calculation of diversity indexes, therefore this will not alter the diversity results.*

*Original L428: However, the presence of the genera acetoclastic Methanosaeta and Methanosarcina, which possess a more diverse genome allowing them to perform hydrogenotrophic, acetoclastic and methylotrophic methanogenesis, suggests that the community holds a metabolic potential to produce $CH_4$ under altered environmental conditions.*

*Authors revisions: However, the presence of the acetoclastic genera Methanosaeta and Methanosarcina, which possess a more diverse genome allowing them to perform hydrogenotrophic, acetoclastic and methylotrophic methanogenesis, suggests that the community holds a metabolic potential to produce $CH_4$ under altered environmental conditions. Furthermore, members of order Methanosarcinales were also detected that hold the functional potential to perform anaerobic oxidation of $CH_4$ and is carried out by anaerobic methane-oxidizing archaea, further increasing the functional potential of the methanogenic community.*

*L114-115 Please clarify whether the "beta-diversity" here refers to both of the CH4 producing and consuming microorganisms. And please explain why such increases is thought to increase with increasing CH4 emission.*

For clarification, we have rewritten this sentence:

*Original: (2) determine whether the $\beta$-diversity increases with increasing $CH_4$ emission*

*Authors revision: (2) determine whether the combined CH$_4$ producing and consuming community β-diversity increases with higher CH$_4$ flux*

*L142 Is the n=6 per mesocosm?*

We have rewritten this sentence to clarify this:

*Original: During the experiment, weekly to bi-weekly (final 3 weeks, n = 6) measurements of CO$_2$ and CH$_4$ fluxes were conducted.*

*Authors revision: During the final three weeks of the experiment, bi-weekly measurements of CO$_2$ and CH$_4$ fluxes were conducted (n = 6 per mesocosm).*

*L166 This reviewer was not able to comprehend the phrase "based of comparison with isotopic mass spectrometer". Please rewrite to clarify.*

We understand that this phrase may be confusing. We have adjusted this section as follows:

*Original: The CH$_4$ emission and its δ$_{13}$C signature were determined using a cavity ring-down laser absorption spectrometer (CRDLAS) with the closed chamber technique described above (G2201i, Picarro, Santa Clara, USA). The surface of each peat mesocosm was covered with a transparent cylindrical chamber for 25-30 minutes while the CH$_4$ mixing ratio and δ$_{13}$C-CH$_4$ was recorded with 1 second intervals. Data was averaged into one minute averages. CH$_4$ emission were calculated using linear fitting, and the δ$_{13}$C signature of emitted CH$_4$ was determined with a Keeling plot intercept approach (Keeling, 1958; Thom et al., 1993). The resulting δ$_{13}$C-CH$_4$ values were corrected by adding a constant value of 3.4 ‰, based of comparison with isotopic mass spectrometer.*

*Authors revision: The CH$_4$ emission and its δ$_{13}$C signature were determined using a cavity ring-down laser absorption spectrometer (CRDLAS) with the closed chamber technique described above (G2201i, Picarro, Santa Clara, USA). The surface of each peat mesocosm was covered with a transparent cylindrical chamber for 25-30 minutes while the CH$_4$ mixing ratio and δ$_{13}$C-CH$_4$ was recorded with 1 second intervals. Data was averaged over one minute and the δ$_{13}$C signature of emitted CH$_4$ was determined with a Keeling plot intercept approach (Keeling, 1958; Thom et al., 1993). We compared values from the CRDLAS instrument with an isotope ratio mass spectrometer (IRMS) by taking air samples from the flux chamber during measurements from the CDRLAS and analyzing these with the IRMS (Rinne et al., 2022). The values from the IRMS indicated a bias of -3.4 ‰ on the CRDLAS, thus we have corrected the values of the δ$_{13}$C signature by adding 3.4 ‰.*

*L179-180 Please provide the access date and/or the version of KEGG database used in this study.*

We have changed the text accordingly:

*Original: Genes encoding enzymes closely related to the $CH_4$ production and oxidation in pathway map00680 were identified from the Kyoto Encyclopedia of Genes and Genomes (KEGG).*

*Authors revision: Genes encoding enzymes closely related to the $CH_4$ production and oxidation in pathway map00680 were identified from the Kyoto Encyclopedia of Genes and Genomes (KEGG), database version 88.*

*L189-190 What is "low TE"? Please explain.*

We have rewritten this sentence to better explain this:

*Original: Depending on the extracted DNA concentration, 150 ng or 1 μg of genomic DNA in a total volume of 100 μl low TE*

*Authors revision: Depending on the extracted DNA concentration, 150 ng or 1 μg of genomic DNA in a total volume of 100 μl low Tris-Ethylenediaminetetraacetic acid buffer (TE buffer).*

*Section 2.7 It is not clearly stated that what data is being used to calculate the Bray-Curtis dissimilarity and the various statistical tests. This makes it a bit difficult to interpret the results.*

We have rewritten this sentence to clarify the data used for these tests:

*Original: Further statistical tests for use on genomic data, including the Permutational multivariate analysis of variance (PERMANOVA), α-diversity and β-diversity, and Nonmetric Multidimensional Scaling (NMDS)*

*Authors revision: absolute abundances for taxonomic and functional sequences from the KEGG ko:00680 metabolism pathway were used as input for the statistical tests including the Permutational multivariate analysis of variance (PERMANOVA), α-diversity and β-diversity, and Nonmetric Multidimensional Scaling (NMDS).*

*L252 Should it be "between" CH4 fluxes, instead of "within"?*

Changed to "among"

*Original: After observing such large variability within CH₄ fluxes*

*Authors revision: After observing such large variability among CH₄ fluxes*

*L271 What does "the flux of CH4 held a positive relationship to $R_{eco}$" actually mean?*

We mean "correlated positively" and have changed the text to clarify this:

*Original: In an attempt to investigate the relationships between carbon fluxes we conducted a correlation test and found that the flux of CH₄ held a positive relationship to $R_{eco}$*

*Authors revision: In an attempt to investigate the relationships between carbon fluxes we conducted a correlation test and found that the flux of CH₄ correlated positively to $R_{eco}$*

*L272-273 Authors explained that GPP is calculated from NEE and Reco (GPP = NEE −Reco). What was the reason for the authors to examine such correlation relationship stated in L272-273?*

We examined this due to the influence of GPP and Reco on available substrate. We have changed the text to make this point more clearly:

*Original: In an attempt to investigate the relationships between carbon fluxes we conducted a correlation test and found that the flux of CH₄ held a positive relationship to $R_{eco}$ ($R_2 = 0.60$, $p \leq 0.04$), but not to GPP or NEE (fig 2). When analysing $CO_2$ fluxes, GPP held a strong negative relationship to $R_{eco}$ ($R_2 = 0.70$, $p \leq 0.002$), while NEE held a strong positive relationship to GPP ($R_2 = 0.82$, $p \leq 0.001$) (fig 2).*

*Authors revision: Previous research has shown that CH₄ flux holds a strong correlation to both GPP and $R_{eco}$, which can influence the availability of CH₄ substrates (Ström et al., 2005). In an attempt to investigate whether the relationships between carbon fluxes matched previous research, we conducted a correlation test and found that the flux of CH₄ held a positive relationship to $R_{eco}$ ($R_2 = 0.60$, $p \leq 0.04$), but not to GPP or NEE (fig 2). When analysing $CO_2$ fluxes, GPP held a strong negative relationship to $R_{eco}$ ($R_2 = 0.70$, $p \leq 0.002$), while NEE held a strong positive relationship to GPP ($R_2 = 0.82$, $p \leq 0.001$) (fig 2).*

*L280 Please add "statistically" before "significant".*

Authors response: This change has been made according to the reviewer's suggestion

*L288-289 Is it possible that the less negative value was contributed to higher CH4 oxidation rate in M2 and M4?*

*Authors response: This is of course possible, as the d13C of the emitted methane reflects both processes involved in methanogenesis and methanotrophy. However, combining the d13C value with the fact that this mesocosm had high methane emission makes it likely that the variation is caused by the methanogenesis, not methanotrophy (e.g. Hornibrook 2009; Rinne et al., 2022). We will include a paragraph on the interpretation of d13C in relation to methane emission in discussion section (see next response).*

*Original: L564-566: Furthermore, the positive correlation between $\delta_{13}C$-$CH_4$ to $CH_4$ emission rate indicates the $CH_4$ emission to be mostly controlled by the trophic status for methanogenesis, rather than methanotrophy (Hornibrook, 2009).*

*Authors revision: Furthermore, the positive correlation between $\delta_{13}C$-$CH_4$ to high $CH_4$ emission rates, especially observed in HFM, indicates that the $CH_4$ emission is mostly controlled by the trophic status for methanogenesis, rather than methanotrophy (Hornibrook, 2009).*

*L290-291 It is not intuitive as to why a relationship between CH4 flux and the Keeling intercept is investigated, and thus, what it meant if there is a significant relationship. To help readers to follow, please explain. Explain the keeling method more clearly, why we use keeling to investigate ch4 fluxes*

*We have revised the text to explain this further:*

*Original: Distinct isotopic signatures of individual mesocosms are shown in fig 3. All mesocosms fell within the range of hydrogenotrophic methanogenesis ($\delta_{13}C = -110‰$ to $-60‰$) (Chanton, 2005; Whiticar, 1999). However, M2 (MFM) and M4 (HFM) indicated a slight tendency towards acetoclastic methanogenesis with less negative isotopic signature ($\delta_{13}C = -60‰$ to $-50‰$), both yielding mid $-60‰$ $\delta_{13}C$ Keeling intercepts. A significant positive correlation ($R_2 = 0.5$, $p \leq 0.001$) and significant relationship also existed between $CH_4$ flux and the Keeling intercept shown in fig 3.*

*Authors revision: Distinct isotopic signatures of individual mesocosms are shown in fig 3. The relationship between d13C and $CH_4$ fluxes can be indicative of the processes controlling the spatial variability of the $CH_4$ emissions (Hornibrook 2009; Rinne et al., 2022). A positive correlation between d13C and CH4 fluxes indicates that the variation is due to the substrate availability for methanogenesis, while a negative correlation is indicative for methanotrophy to*

*be the dominant cause for the variability of CH$_4$ flux. All mesocosms fell within the range of hydrogenotrophic methanogenesis (δ13C = −110‰ to −60‰) (Chanton, 2005; Whiticar, 1999) and held a significant positive correlation (R2 = 0.5, p ≤ 0.001), indicating the dominant methanogenesis pathway to be hydrogenotrophic. However, M2 (MFM) and M4 (HFM) indicated a slight tendency towards acetoclastic methanogenesis with less negative isotopic signatures (δ13C = -60‰ to -50‰), both yielding mid -60‰ δ13C Keeling intercepts.*

*Figure 3. There are two apparent groups of Keeling intercepts in MFM. Is there any meaning to it? Also, there is a single LFM data point (orange at CH4 flux of ~260 umol m-2 h-1) appearing amidst of the MFM, any explanation why this LFM gave a higher CH4 flux compared to other 5 LFM datapoints? should this datapoint be omitted from the analysis?*

*Authors response: The division of mesocosms to LFM, MFM and HFM groups was based on their average methane emission rates while Figure 3 shows the individual measurements of methane emissions. At one time, the emission from LFM and MFM mesocosms was high, leading to the data point mentioned moving away from the other data points. As this study is focused on a replicated peak growing season, and not a temporal scale, we prefer not to remove the data point just because it is an outlier.*

*L298-299 What unit is it? phyla OR OTU OR genera as in L307? (Add and genus level)*

*We have changed this sentence to clarify this:*

Original: In total, 20 methanogenic Archaea and 5 methanotrophic Bacteria were detected.

Authors response: In total, 20 genera of methanogenic Archaea and 5 methanotrophic Bacteria were detected.

*L308 It would be clearer to say "methanogenic community" (provided that ANMEs are not detected), instead of "proportion.*

Authors response: This change has been made according to the reviewer's suggestion

*L315 It should be "CH4 oxidizing"*

*Authors response: This change has been made according to the reviewer's suggestion*

*L317 Alphaproteobacteria is at the class level (!)*

*Added this information:*

*Original: 5 genera of CH₄ reducing Bacteria were detected including methanotrophs from Alphaproteobacteria, Gammaproteobacteria and Verrucomicrobia class.*

*Authors response: 5 genera of CH₄ reducing Bacteria were detected including methanotrophs from class level Alphaproteobacteria, Gammaproteobacteria and Verrucomicrobia.*

L327-329 Such statement is not meaningful in statistics.

*Authors response: This sentence has been removed according to the reviewer's suggestion*

L344 the second and third highest "dissimilarity"
Table 1-6 Please explain to the readers how to understand the p-value.
Perhaps missing "in", in average "in" MFM and HFM?

*Author response: The explanation of the p-value is described in the table text "p-value of the permutation test (Probability of getting a larger or equal average contribution in random permutation of the group factor)"*

L369 "CH4 metabolism (PATH: KO00680) made up 17% of the captured genes" ... this is confusing because this reviewer learned from the earlier text that "captured metagenomics" data targeted only "the CH4 production and oxidation in pathway map00680" by using the 193,386 individual designed probes. (explain about off target hybridization)

*Authors response: The method of captured metagenomics allows the user to target high number of genes sequences, but as with any method it is not perfect. Off target sequences are known to hybridize to the custom designed probes if they hold a high enough similarity to the binding site. Therefore, we filter out the off target sequences using bioinformatics. One such way is by using the MG-RAST pathway filter, i.e. path 00680 for the CH4 cycle.*

L382 How should one understand the term "cumulative sum"? Please clarify and provide guidance to readers.

*We have changed the text to explain this:*

*Original: In total, 21 genes of the 109 contributed to 70% of the cumulative sum (table 4, 5 and 6).*

*Authors response: In total, 21 genes of the 109 contributed to 70% of the cumulative sum, i.e. the contributions for each gene in descending order (table 4, 5 and 6).*

*L382-405 It was not easy to follow the comparisons and the results are very similar in the three comparisons.*

Authors response: Yes, the results for the three comparisons are very similar and thus difficult to interpret for a clear take home message. Throughout the passage of text, we have referred to tables, figures and statistics to clarify the reader. We believe this further reinforces our conclusion that the functional potential of the methane producing and consuming community displays minor importance in explaining spatial variability of CH₄ fluxes.

*Discussion When referring to specific results obtained in this study, please cite the corresponding figures/tables. This is helpful for readers to follow and evaluate the arguments.*

Authors response: This change has been made according to the reviewer's suggestion, please find references to tables and figures now within the discussion.

*L431-432 Error: "Proteobacteria" should be before "and"*

Authors response: This change has been made according to the reviewer's suggestion

*L435-436 It is not clear what "observed pathways" are being referred to...As stated here, d13C suggested dominant methane production pathway but d13C does not inform consumption pathways. Genomic information tells only the metabolic potential.*

We understand that this sentence may be misinterpreted, which is why we have rewritten it as follows:

Original: we can expect CH₄ production and consumption to still occur, but possibly using alternative metabolic pathways than currently observed

Authors revision: our results indicate that we can expect CH₄ production and consumption to still occur, as the community holds the functional potential to continue producing or reducing CH₄, possibly using alternative metabolic pathways such as acetoclastic or methylotrophic methanogenesis.

*L441-442 Please provide information about "the absence of acetogenesis and fermentation"...then it would be helpful for readers to relate the following statement "the less dominant functional...." at their study site.*

Information added.

*Original: In the absence of acetogenesis and fermentation, the less dominant functional groups (i.e. acetoclastic and methylotrophic methanogens) may still remain dormant, due to the absence of necessary substrates to metabolize.*

*Authors revision: In the absence of acetogenesis and fermentation, that produce the necessary products for acetoclastic and methylotrophic methanogenesis, the less dominant acetoclastic and methylotrophic methanogens may still remain dormant due to the absence of necessary substrates to metabolize.*

**L445 The "spatial" info of the highly variable CH4 flux is not given, and it would be good for readers to know the spatial variability represented by M1-M9.**

*Authors response: We have added a reference to figure 1 in this sentence. The variability within and across different mesocosms is apparent from the boxplots.*

*Original: We observed a high spatial variability in $CH_4$ flux, which is consistent with research conducted in other temperate peatlands*

*Authors revision: We observed a high spatial variability in $CH_4$ fluxes (fig 1), which is consistent with research conducted in other temperate peatlands*

**L449-450 $R_{eco}$ was measured for the mesocosm, meaning that the high respiration was a result of the whole community. This reviewer considers that it is inappropriate to use captured metagenomes (targeting methane cycling community) to explain an observation coming from the whole community. Therefore, this statement is considered weak or even misleading.**

*We agree with the reviewer that Reco is the result of a much larger community than just the methane cycling community, and it also includes autotrophic respiration from the plants. We have weakened this statement as follows:*

*Original: One potential reason for the high respiration from HFM could be the significantly higher relative abundance of pmoA. The pmoA gene codes for the first step in methanotrophy, where CH4 is reduced to methanol, and finally CO2, which is often used as a proxy for methanotrophy (Franchini et al., 2015; Freitag et al., 2010). The higher abundance of pmoA may indicate a higher rate of methanotrophy, which may help to explain the higher CO2 flux respired by the methanotrophs in HFM. In addition, higher plant productivity causes higher autotrophic respiration, which generally makes up ~50% of Reco. However, the vegetation may also be supplying more substrates to the microbial community, which in turn is consumed and respired in the form of CO2.*

*Authors revision: The high respiration from HFM coincides with a significantly higher relative abundance of pmoA. The pmoA gene codes for the first step in methanotrophy, where $CH_4$ is reduced to methanol, and finally $CO_2$, which is often used as a proxy for methanotrophy (Franchini et al., 2015; Freitag et al., 2010). The higher abundance of pmoA may indicate a higher rate of methanotrophy, which would contribute to higher respiration. However, as we have used a targeted approach, we cannot conclude that the pmoA gene is significantly higher than other carbon reducing genes outside of methanotrophy. Moreover, autotrophic respiration from plants can be as large as heterotrophic respiration in peatlands, which further complicates this picture (see e.g. Järveoja et al. 2020). Thus, we can only conclude that methanotrophy can contribute to higher Reco but it is not the only contributor.*

**L492 Methylocella is a close relative of high-affinity methanotrophs (upland soil cluster alpha). Would any of the detected Methylocella data be coming from high-affinity methanotrophs?**

*Author response: We can of course not guarantee that all taxonomic assignments are 100% correct all the time, however, there appears there are enough high-affinity methanogen sequences that the classifications would hold. Also, do we know what the ppm levels of methane would be in our samples? High-affinity methanotrophs may become a factor if the methane has very low. Regardless, since we do not see any significant change in the abundance between the groups, I suggest that the taxonomic assignments are trustworthy.*

**L506 Choice of word. Should not use "were".**

*Changed accordingly.*

*Original: contrary to results found by Zhang et al. (2019) were the authors observed significant correlation between mcrA and $CH_4$ flux.*

*Authors revision: contrary to results found by Zhang et al. (2019) where the authors observed significant correlation between mcrA and $CH_4$ flux.*

**L532 Depending the database used for gene annotation, CODH is a symnonym for carbon monoxide dehydrogenase. CODH/ACS is being used by many if not all methanogens in the reductive acetyl-coA pathway for CO2 fixation, so it is not surprising to see that in the results. And though hdr and CODH genes do not directly involved in methane producing pathway, they are essential for the living of methanogens.**

*Authors response: I believe you have written it very clearly and we agree with your statement. We have added this information to the text.*

*Original: These genes code for CO dehydrogenase and are involved in the Acetyl-CoA pathway, which is not directly included in methanogenesis.*

*Authors revision: These genes code for CO dehydrogenase and are involved in the Acetyl-CoA pathway, which is not directly involved in methane producing pathway, but are essential for the living of methanogens, therefore it is expected to observe CO dehydrogenase genes in high abundance.*

**L548 "our" is likely a typo.**

*The typo has been removed.*

**L553 The word "indicating" is too strong. And please be more specific to say the microbial group, and not just "microbes". (Suggesting)**

*Changed to 'suggesting'*

*Original: HFM held the highest dissimilarity indicating that as the $CH_4$ flux increases, the abundance and variability of microbe's increase.*

*Authors revision: HFM held the highest dissimilarity, suggesting that as the $CH_4$ flux increases, the abundance and variability of microbial group.*

**L566 The discussion will benefits if authors further elaborate on what they think about the trophic status of methanogenesis in HFM, MFM and LFM.**

*Authors response: we have added additional discussion material following L566.*

*Authors revision: Covariation of d13C-CH4 with methane emission rates suggests that the spatial variation in methane emissions are determined largely by the variations in the precursor availability and thus trophic status within a mire (Rinne et al., 2022). Similarly, within the mesocosms studied here, the d13C-CH4 correlates positively with methane emission rates between the mesocosms and flux categories, indicating the trophic status to exert major control on this variation.*

**L576 Is it right that it is over 50% of the methane-cycling community? Please clarify.**

*Authors response: Yes, this statement is correct. For clarity we have added that it is 50% of the methane producing and consuming community.*

*Original: The dominant methanogen, Methanoregula, made up over 50% of the community composition.*

*Authors revision: The dominant methanogen, Methanoregula, made up over 50% of the methane producing and consuming community composition.*